REPLICATION STUDY

# Replication Study: Intestinal inflammation targets cancer-inducing activity of the microbiota

Kathryn Eaton, Ali Pirani, Evan S Snitkin, Reproducibility Project: Cancer Biology*

Department of Microbiology and Immunology, University of Michigan Medical School, Ann Arbor, United States

**Abstract** As part of the Reproducibility Project: Cancer Biology we published a Registered Report (Eaton et al., 2015) that described how we intended to replicate selected experiments from the paper "Intestinal Inflammation Targets Cancer-Inducing Activity of the Microbiota" (Arthur et al., 2012). Here we report the results. We observed no impact on bacterial growth or colonization capacity when the polyketide synthase (*pks*) genotoxic island was deleted from *E. coli* NC101, similar to the original study (Supplementary Figure 7; Arthur et al., 2012). However, for the experiment that compared inflammation, invasion, and neoplasia in azoxymethane (AOM)-treated interleukin-10-deficient mice mono-associated with NC101 or NC101Δ *pks* the experimental timing of the replication attempt was longer than that of the original study. This difference was because in the original study the methodology was not clearly stated and likely led to the increased mortality and severity of inflammation observed in this replication attempt. Additionally, early death occurred during AOM treatment with higher mortality observed in NC101Δ *pks* mono-associated mice compared to NC101, which was in the same direction, but more severe than the original study (Suppleme1ntal Figure 10; Arthur et al., 2012). A meta-analysis suggests that mice mono-associated with NC101Δ *pks* have higher mortality compared to NC101. While these data were unable to address whether, under the conditions of the original study, NC101 and NC101Δ *pks* differ in inflammation, invasion, and neoplasia this replication attempt demonstrates that clear description of experimental methods is essential to ensure accurate reproduction of experimental studies.
DOI: https://doi.org/10.7554/eLife.34364.001

*For correspondence:
tim@cos.io;
nicole@scienceexchange.com

Group author details:
Reproducibility Project: Cancer Biology See page 15

## Introduction

The Reproducibility Project: Cancer Biology (RP:CB) is a collaboration between the Center for Open Science and Science Exchange that seeks to address concerns about reproducibility in scientific research by conducting replications of selected experiments from a number of high-profile papers in the field of cancer biology (Errington et al., 2014). For each of these papers a Registered Report detailing the proposed experimental designs and protocols for the replications was peer reviewed and published prior to data collection. The present paper is a Replication Study that reports the results of the replication experiments detailed in the Registered Report (Eaton et al., 2015) for a 2012 paper by Arthur et al., and uses a number of approaches to compare the outcomes of the original experiments and the replications.

In 2012, Arthur et al. reported results that intestinal inflammation modifies the gut microbiota affecting the progression of colorectal cancer (CRC). The model used in that study was one of a group of related models that are commonly used to study the role of inflammation in colon carcinogenesis (Kanneganti et al., 2011). These models use a combination of treatment with azoxymethane (AOM), a proximate carcinogen, with an initiator of local inflammation. The group of models vary as to the dose and duration of AOM treatment as well as the treatment used to induce inflammation.

Inflammatory insults used may be chemical, most commonly dextran sodium sulfate (DSS, a local irritant), genetic (e.g. engineered absence11 of a regulator of inflammation, such as $Il10^{-/-}$), infectious (e.g. *Helicobacter hepaticus*, *Escherichia coli* (*E. coli*), or *Salmonella typhimurium*), or a combination of these (for details, see *Kanneganti et al., 2011*). The model used by Arthur and colleagues was a combination of interleukin-10-deficient ($Il10^{-/-}$) mice and *E. coli* mono-association to produce a background of chronic inflammation followed by six weekly injections of AOM to induce neoplastic transformation. Using this inflammation-induced CRC model, Arthur and colleagues reported that germ-free mice mono-associated with the commensal mouse adherent-invasive *E. coli* strain NC101 developed invasive mucinous carcinomas which did not occur in mice mono-associated with *Enterococcus faecalis*, another colitis-inducing bacterial strain (*Arthur et al., 2012*). NC101 harbors the polyketide synthase (*pks*) pathogenicity island that encodes the biosynthetic machinery for synthesizing the genotoxin colibactin (*Nougayrède et al., 2006*). Mono-association of NC101 led to enhanced tumor multiplicity and invasion in AOM-treated germ-free $Il10^{-/-}$ mice, which was decreased in AOM-treated germ-free $Il10^{-/-}$ mice mono-associated with an isogenic mutant deficient for *pks* island (NC101∆ *pks*), without altering colonic inflammation (*Arthur et al., 2012*).

The Registered Report for the 2012 paper by Arthur et al. described the experiments to be replicated (Figure 4A–F, and Supplemental Figure 7 and 10), and summarized the current evidence for these findings (*Eaton et al., 2015*). Since that publication there have been additional studies investigating the effect of *pks*-harboring *E. coli* strains to enhance tumorigenesis. Similar observations were observed using $APC^{Min/+}$ mice (*Bonnet et al., 2014*), germ-free $APC^{Min/+}$; $Il10^{-/-}$ mice (*Tomkovich et al., 2017*), or a AOM-DDS xenograft mouse model of CRC (*Cougnoux et al., 2014*). A follow-up study by Arthur and colleagues, reported that colonic inflammation was necessary for the tumor-promoting activity of NC101 through modulation of specific microbial genes (*Arthur et al., 2014*).

The outcome measures reported in this Replication Study will be aggregated with those from the other Replication Studies to create a dataset that will be examined to provide evidence about reproducibility of cancer biology research, and to identify factors that influence reproducibility more generally.

## Results and discussion

### Impact of *pks* island deletion on bacterial growth

Using the same commensal mouse adherent-invasive *E. coli* NC101 strain and an isogenic *pks*-deficient (NC101∆ *pks*) strain as the original study (*Arthur et al., 2012*), we confirmed deletion of the *pks* island by PCR and whole genome sequencing, which revealed no variants or insertions/deletions other than the desired *pks* deletion between the two isogenic strains (*Figure 1—figure supplement 1*). The two bacterial strains were analyzed to determine if the absence of *pks* affected bacterial growth. This is comparable to what was reported in Supplemental Figure 7 of *Arthur et al. (2012)* and described in Protocol 1 in the Registered Report (*Eaton et al., 2015*). Similar to the original study, NC101 and NC101∆ *pks* growth curves were visually equivalent to each other (*Figure 1*, *Figure 1—figure supplement 2*), indicating deletion of *pks* does not affect *E. coli* growth in vitro. The intrinsic doubling time of the NC101 strain was 36 min, 95% CI [27-45], while the doubling time of the NC101∆ *pks* strain was 52 min, 95% CI [6-97]. This compares to the original study that had an estimated doubling time of ~53 min for the NC101 strain and ~64 min for the NC101∆ *pks* strain. To summarize, for this experiment we found results that were similar to the original study.

### Intestinal tumorigenesis and inflammation of germ-free $Il10^{-/-}$ mice mono-associated with *E. coli* NC101 or NC101 ∆*pks*

AOM-colitis models of carcinogenesis have been comprehensively reviewed (*Kanneganti et al., 2011*); however, it is important to note the unique features of the model used in the original study and this replication attempt. First, the model used germ-free mice. Germ-free mice differ from conventional mice in that first, they do not normally develop colitis, even in the absence of IL-10 (*Eaton et al., 2011*). $Il10^{-/-}$ mice that are housed in the presence of enteric microbes develop varying

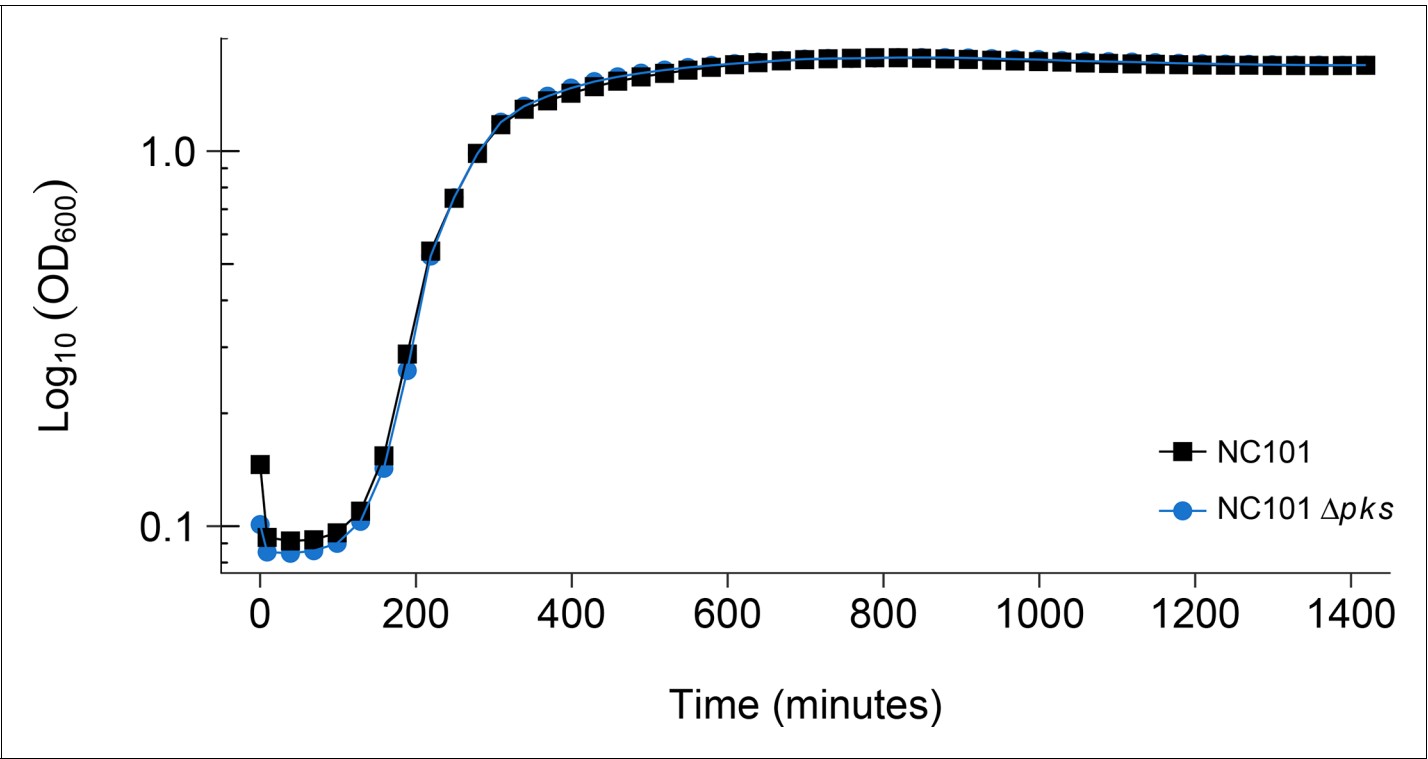

**Figure 1.** Detection of *pks* island and impact of *pks* island deletion on bacterial growth in vitro. In vitro growth curve of *E. coli* NC101 and *E. coli* NC101Δ *pks*. Overnight bacterial cultures were diluted 1:500 in Luria-Bertani (LB) broth and incubated at 37°C. Absorbance at 600 nm (OD$_{600}$) was measured every 30 min. Representative growth curves of 3 independent biological repeats. The intrinsic doubling time was determined to be 36 min, 95% CI [27-45] for the NC101 strain and 52 min, 95% CI [6-97] for the NC101Δ *pks* strain. Additional details for this experiment can be found at https://osf.io/54rgt/.

DOI: https://doi.org/10.7554/eLife.34364.002

The following figure supplements are available for figure 1:

**Figure supplement 1.** Confirmation of *pks* island deletion.

DOI: https://doi.org/10.7554/eLife.34364.003

**Figure supplement 2.** Additional dilutions used with growth curve assay.

DOI: https://doi.org/10.7554/eLife.34364.004

levels of colitis that appear to depend on their specific microbiota. Germ-free mice have been used fairly extensively in a related AOM-DSS model of carcinogenesis, but rarely in AOM-infection models. Also, germ-free mice differ from specific-pathogen-free (SPF) mice in their hepatic metabolism. Because AOM is a hepatotoxin in addition to a carcinogen, its toxicity may be altered in mice without a normal microbiota (*Selwyn et al., 2016*; *Tung et al., 2017*). The second unique aspect of the current model is that it uses mutant mice on a 129 SvEv background. Mouse strains differ in their response to AOM with the effect also modulated by non-genetic factors, like diet, which tend to vary widely across studies making it difficult to directly compare results (*Bissahoyo et al., 2005*) For these reasons, we attempted to control both genetic and non-genetic factors between the original study and this replication attempt.

To test whether absence of the *pks* island reduced tumorigenic potential, but not inflammatory potential of *E. coli*, we attempted to independently replicate an experiment similar to the one reported in Figure 4A–F, and Supplemental Figure 10, of *Arthur et al. (2012)*. The protocol used was described in Protocol 3 in the Registered Report (*Eaton et al., 2015*), which was based on the available information provided in the original published paper (*Arthur et al., 2012*) and through communication with the authors of the original study. Germ-free *Il10*[-/-] mice on a 129/SvEV background (derived from the same germ-free colony used in the original study) were mono-associated with either the NC101 or NC101Δ *pks* isolate described above, treated with AOM, and then assessed for colonic inflammation and tumorigenesis. The original study also reported cohorts of

mice that did not receive AOM treatment (assessed at 12 weeks), or received AOM treatment and were assessed at 14 weeks. These additional cohorts were not included in the design of this replication attempt. Furthermore, the number of mice required for this study to have sufficient power to detect the originally reported effect sizes were determined *a priori* and took into account the number of anticipated animal deaths that would occur prior to the 18 week assessment based on the originally reported survival rates. To summarize, bacterial colonization was confirmed 4 weeks after mono-association, after which AOM was administered weekly for a total of 6 injections, and then mice were monitored for 18 weeks after the last AOM injection (experimental timing visualized in *Figure 2A*). The timing for this experiment was based on information available in the original paper (*Arthur et al., 2012*) and remained after informal review and feedback by the authors of the original paper during preparation of the Registered Report manuscript, peer review of the Registered Report, and post-publication peer review of the published Registered Report. During peer review of this Replication Study, however, one of the reviewers suggested that the experimental timing used in this replication attempt was different from the original study, which evaluated colonic inflammation and tumorigenesis 18 weeks after mono-association rather than 18 weeks after AOM treatment as was described in the Registered Report and performed in this study. Thus, this replication attempt had an experimental endpoint 9 weeks longer than the original study. This methodological error, which confounds the interpretation of the results of this replication attempt, was based on the methods derived from the original study and not corrected on review of the Registered Report (*Eaton et al., 2015*). Others have reported how assumptions in experimental timing or methods hindered their efforts to understand how seemingly similar experiments produced different results (*Hines et al., 2014*; *Lithgow et al., 2017*). One approach to mitigate the potential for misinterpreting complex study designs is to include a timeline diagram or flowchart as recommended by the ARRIVE Guidelines (*Kilkenny et al., 2010*).

In this study, we found most mice did not survive to the planned 18-week post-AOM time point, and either died or were euthanized prior to the intended study end-point (*Figure 2A*). Despite efforts to include more animals into the study (n=39 for NC101 mono-associated mice; n=45 for NC101Δ *pks* mono-associated mice), we did not achieve the planned number of animals at the end point of 18 weeks post-AOM treatment. Only 11 mice survived to 17-18 weeks after AOM treatment (4 mono-associated with NC101; 7 mono-associated with NC101Δ *pks*), while a total of 25 mice survived 14 weeks post-AOM treatment or later. During the course of the entire study, there was a median survival of 154 days (range: 31-176 days) and 57 days (range: 29-188 days) for NC101 and NC101Δ *pks* mono-associated mice, respectively, which was not statistically significant (log-rank (Mantel-Cox) test; *p* = 0.0608). Regardless of mono-associated bacterial strain, mortality was highest during AOM treatment; however during this interval, mortality was greater in mice mono-associated with NC101Δ *pks* compared to mice mono-associated with NC101. In the NC101Δ *pks* group, fewer than half the animals survived beyond the last AOM treatment. Importantly, there were similar levels of colonization capacity *in vivo* for both strains of bacteria (*Figure 2—figure supplement 1*), similar to the original study, suggesting bacterial load was not a factor in the survival differences.

Interestingly, while a similar distribution of male and female mice were assigned to both strains for mono-association (NC101: female=21, male=18; NC101Δ *pks*: female=23, male=22), male mice became overrepresented during the course of this study: 14 weeks post-AOM treatment (NC101: female=5, male=9; NC101Δ *pks*: female=3, male=8); 17-18 weeks post-AOM treatment (NC101: female=0, male=4; NC101Δ *pks*: female=2, male=5). In the original study 4 female and 8 male mice were mono-associated with NC101 and 8 male mice were mono-associated with NC101Δ *pks* (Arthur, personal communication) of which 14 mice (9 mono-associated with NC101; 5 mono-associated with NC101Δ *pks*) survived to 18 weeks after mono-association (*Arthur et al., 2012*); however, the sex distribution of the surviving mice was not published or communicated.

To facilitate a direct comparison of these results to the original study we determined survival up to 18 weeks following mono-association, similar to the timing performed in the original study (*Figure 2B*). When treating 18 weeks following mono-association as the endpoint (i.e. ignoring events after this time point), we observed the absence of the *pks* island had an impact on survival (NC101: 53.8%; NC101Δ *pks*: 26.7%). This corresponds to a median survival of 57 days for NC101Δ *pks* while the median survival for NC101 mono-associated mice could not be determined since more than half of the animals were still alive at 18 weeks following mono-association. An exploratory analysis to compare the survival distributions between the two groups during this timeframe was

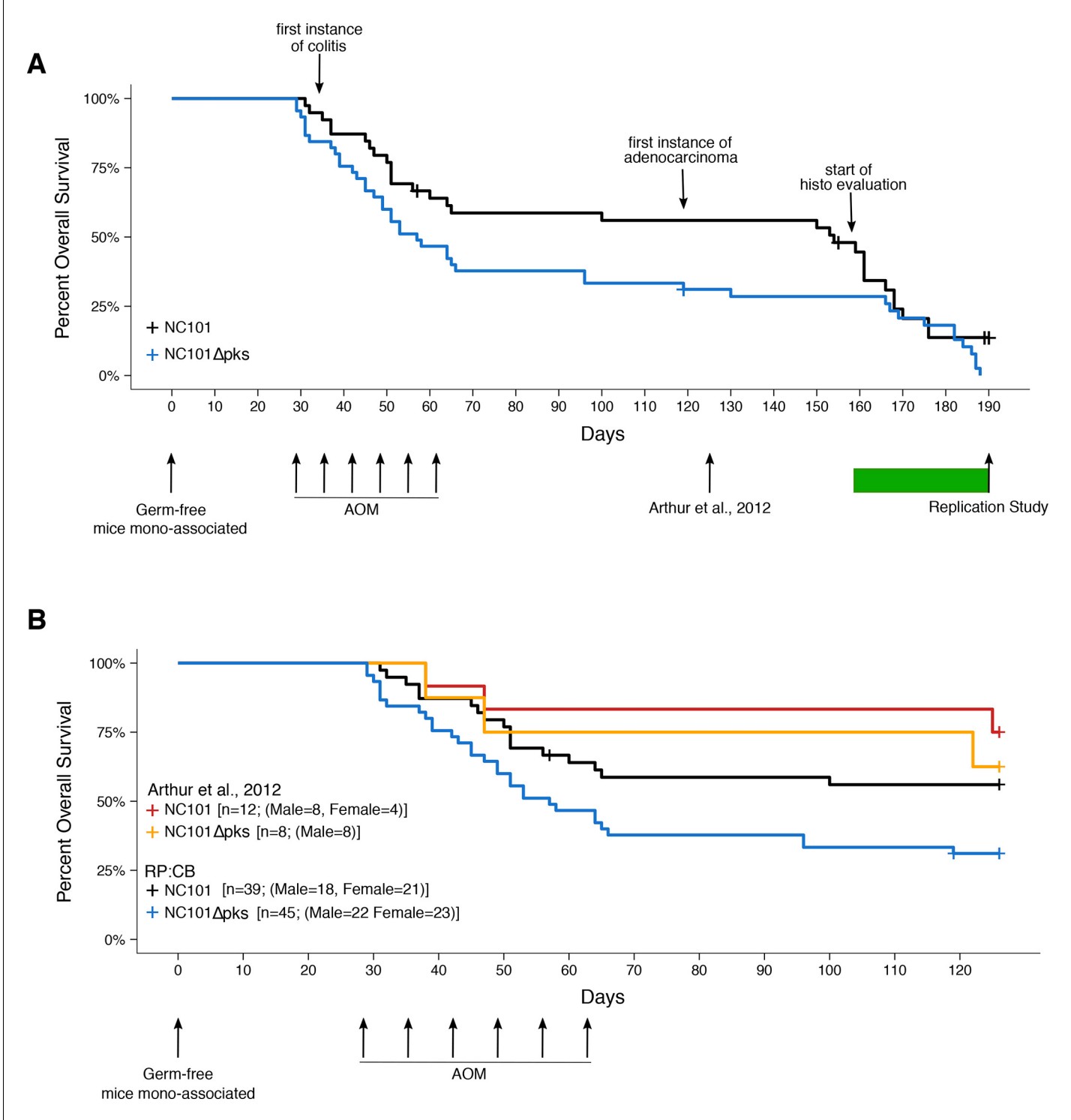

**Figure 2.** Impact of *pks* island deletion on mouse survival. Female and male *Il10*[-/-] germ-free mice were randomly assigned to be mono-associated with either *E. coli* NC101 or *E. coli* NC101Δ *pks* at age 7-12 weeks. Four weeks after mono-association, mice received six weekly azoxymethane (AOM) injections to induce colon tumors. Mice were monitored until euthanized due to health complications or the pre-specified study end-point of 18 weeks after the last AOM injection (Replication Study). This experimental end-point was longer than the original study, which euthanized mice at 18 weeks after mono-association (*Arthur et al., 2012*). Number of mice monitored: n=39 for NC101 mono-associated mice and n=45 for NC101Δ *pks* mono-associated mice. Early in the study a few isocages were contaminated and had to be removed from the study. This did not affect the other isocages which remained gnotobiotic. Animals where bacterial contamination was detected (n=7 in 3 cages) were censored in the plots (denoted by a cross line). (A) Kaplan-Meier plot of overall survival starting at time of mono-association until 18 weeks after the last AOM injection, which is the experimental

*Figure 2 continued on next page*

*Figure 2 continued*

timing used in this replication. Green bar indicates when euthanized mice were histopathologically (histo) evaluated for inflammation and tumorigenesis. Log-rank (Mantel-Cox) test of NC101Δ *pks* compared to NC101 (*p* = 0.0608); HR=1.57, 95% CI [0.98, 2.51]. (**B**) Kaplan-Meier plot of overall survival starting at time of mono-association until 18 weeks after mono-association, which is the experimental timing used in the original study (***Arthur et al., 2012***). Original and replication data are plotted for direct comparison. Exploratory analysis of replication data: Log-rank (Mantel-Cox) test of NC101Δ *pks* compared to NC101 (*p* = 0.0230); HR=1.95, 95% CI [1.10, 3.45]. Additional details for this experiment can be found at https://osf.io/pm5xa/.

DOI: https://doi.org/10.7554/eLife.34364.005

The following figure supplements are available for figure 2:

**Figure supplement 1.** Impact of *pks* island deletion on bacterial growth *in vivo*.

DOI: https://doi.org/10.7554/eLife.34364.006

**Figure supplement 2.** Histopathology of mouse tissues.

DOI: https://doi.org/10.7554/eLife.34364.007

statistically significant (log-rank (Mantel-Cox) test; *p* = 0.0230). The original study reported that the absence of the *pks* island had a small, but not statistically significant effect on mouse survival (NC101: 75% (9 of 12 mice); NC101Δ *pks*: 62.5% (5 of 8 mice)) (***Arthur et al., 2012***). When considering only the survival data up to 18 weeks after mono-association, which is the timing performed in the original study, we found results that were in the same direction as the original study.

During this study, early death that occurred during AOM treatment was associated with widespread liver lesions. This outcome was not reported in the original study; however, the different outcomes between the two studies could be due to methodological details that were unaccounted for (***Bramhall et al., 2015***), or other unknown differences between the two studies. AOM is a known hepatoxin, but the dose and protocol used were below the published toxic dose for AOM in mice (***Bissahoyo et al., 2005***), and were not expected to cause lesions. Four mice that were found dead 1-2 days following the first or second AOM treatment (NC101 = 1, NC101Δ *pks* = 3) had massive acute hepatic necrosis that appeared severe enough to have caused death (***Figure 2—figure supplement 2A***). The other mice that died during AOM treatment had liver lesions of less severity, including widespread hepatocellular vacuolation and multifocal hepatocellular necrosis (***Figure 2—figure supplement 2B***). Mice that died after the AOM treatment phase all had chronic liver lesions suggestive of a regenerative response to ongoing damage. These included anisocytosis often with giant hepatocytes well beyond what would be expected in aged mice, atrophy and disorganization of hepatic acini, and multifocal single-cell necrosis (***Figure 2—figure supplement 2C***). Surprisingly, there was no evidence of fibrosis even in the most chronic lesions. Although the previous, or ongoing, liver damage likely contributed to the animals poor condition in mice that survived the AOM treatment, the chronic liver lesions did not appear severe enough to have caused death directly.

In the mice that died or were euthanized prematurely, colitis first appeared at the start of AOM treatment (5 weeks after mono-association) and was present in all mice that were examined histologically 9 weeks or more after mono-association. Most mice also had typhlitis, which was less severe than the colitis. Inclusion of AOM-only controls could have indicated any increased susceptibility of the mice used in this study to AOM and should be considered in the experimental design of future studies.

Colon adenocarcinoma was first detected in mice that died 8 weeks after AOM treatment (17 weeks after mono-association) and was present in all mice euthanized 13 weeks or more after AOM treatment. Notably, a few mice died at time points close to the time that mice were harvested in the original study: one with typhlitis (NC101 at 14 weeks), two with severe colitis (NC101Δ *pks* at 14 weeks; NC101Δ *pks* at 14 weeks), one with typhlitis and dysplasia (NC101Δ *pks* at 19 weeks), and three with colon adenocarcinoma (NC101Δ *pks* at 17 weeks; NC101 at 22 weeks; NC101 at 22 weeks). Furthermore, five mice (between 19 and 27 weeks after mono-association) had anal squamous cell carcinoma in addition to colon adenocarcinoma (NC101 = 3, NC101Δ *pks* = 2) (***Figure 2—figure supplement 2D,E***).

In addition to the increased early death rate during and immediately after AOM treatment in this replication attempt, severity of chronic lesions and extent of neoplasia were greater than what was reported in the original study. This is most likely due to the longer experimental timing that occurred in this replication attempted compared to the original study. Lesions were similar in morphology to

those described in the original study, but more severe. Grossly evident colon thickening was present in mice examined 22 or more weeks after mono-associations. These lesions were widespread and coalescing, and unlike in the original study, individual masses could not be distinguished either grossly or histologically (*Figure 3—figure supplement 1*). This morphology is typical of both neoplastic and non-neoplastic inflammation-associated proliferative lesions in mice (*Boivin et al., 2003*; *Washington et al., 2013*). The most severely affected mice had markedly irregular colon mucosa, sometimes extending from the anus to the mid-proximal part of the colon. The proximal colon and cecum were grossly normal, while the distal half to third of the colon had lesions. Lesions were also sometimes present in the mid-colon, but only when the distal colon was severely affected. These observations were the same whether the mouse was mono-associated with NC101 or NC101Δ *pks*. In the few cases (10 of 24 mice) where histologically detectable non-neoplastic tissue was present between neoplastic foci, an attempt was made to enumerate individual tumors in histologic sections (*Figure 3A*). The median count in sections in which individual tumors were detectable for NC101 mono-associated mice was 3.5, which was greater than in NC101Δ *pks* mono-associated mice (median count of 2). Because individual tumors could not be distinguished, tumor counts cannot be directly compared to the original study where the absence of the *pks* island resulted in a statistically significant decrease in macroscopic tumor burden (estimated median count: NC101 = 8, NC101Δ *pks* = 2), without an impact on tumor size (*Arthur et al., 2012*).

Colonic inflammation and tumorigenesis were scored as far as possible using the same scoring criteria as the original study (*Arthur et al., 2012*) while taking into account other published criteria (see Materials and methods section; *Boivin et al., 2003*; *Rath et al., 1996*; *Washington et al., 2013*). We found that there were no substantial differences in the inflammation, invasion, or neoplasia scores between mice mono-associated with NC101 or NC101Δ *pks*. When considering all the mice that survived 14 weeks post-AOM treatment (5 weeks beyond the endpoint in *Arthur et al., 2012*), or later, the median scores for each measure were at or near the maximum possible for both cohorts (inflammation = 6; invasion = 8; neoplasia = 5) (*Figure 3B–D*). These results are confounded by the increased severity of inflammation and mortality of animals, most likely due to the experimental timing that occurred in this replication, which was longer than what occurred in the original study. Thus, these results cannot be directly compared to the original study which reported that the absence of the *pks* island resulted in a statistically significant decrease in neoplasia scores (NC101: median = 4, range = 4-5; NC101Δ *pks*: median = 4, range = 3-4) and invasion scores (NC101: median = 2, range = 1-6; NC101Δ *pks*: median = 1, range = 0-2), but not inflammation scores (NC101: median = 4, range = 4-4; NC101Δ *pks*: median = 4, range = 3-4) 18 weeks after mono-association (*Arthur et al., 2012*). Although the original study analyzed the ordinal scoring data as interval measurements (by *t* test), which is not appropriate since the mean cannot be defined (*Baker et al., 2014*; *Gibson-Corley et al., 2013*), similar results were obtained when a non-parametric test (i.e. Mann Whitney test) was applied on the original data (inflammation: $U = 27$, $p = 0.233$; invasion: $U = 39.5$, $p = 0.0240$; neoplasia: $U = 37.5$, $p = 0.0297$).

As noted above, the difference in severity of inflammation, invasion, and neoplasia between the two studies are most likely explained by the increased experimental timing that occurred in this replication attempt that differed from the original study. The absolute scores were greater in this replication attempt compared to the original study, particularly for inflammation and invasion, which, combined with the survival and histopathological observations described above suggests that lesions were more severe and/or progressive over time in this replication attempt than in the original study. Any differences attributable to the absence of *pks* would have been masked by the greatly increased severity of the lesions. Additionally, over time it is possible the products of excessive inflammatory responses (e.g. reactive oxygen species), which promote tumorigenesis, become more important than *pks* status. Thus, it is possible that the experimental timing in this mouse model is crucial for differentiating the outcomes of NC101 and NC101Δ *pks*. While subjective differences in histologic interpretation could also account for differences between the studies (*Cross, 1998*; *Gibson-Corley et al., 2013*), the level of variation observed between these two studies is likely greater than would be expected due to differences in interpretation alone. To summarize, since this replication attempt did not model the kinetics of the mouse model as they occurred in the original study, these data are unable to address whether, under the conditions of the original study, NC101 and NC101Δ *pks* differ in inflammation, invasion, and neoplasia. These results highlight the importance of

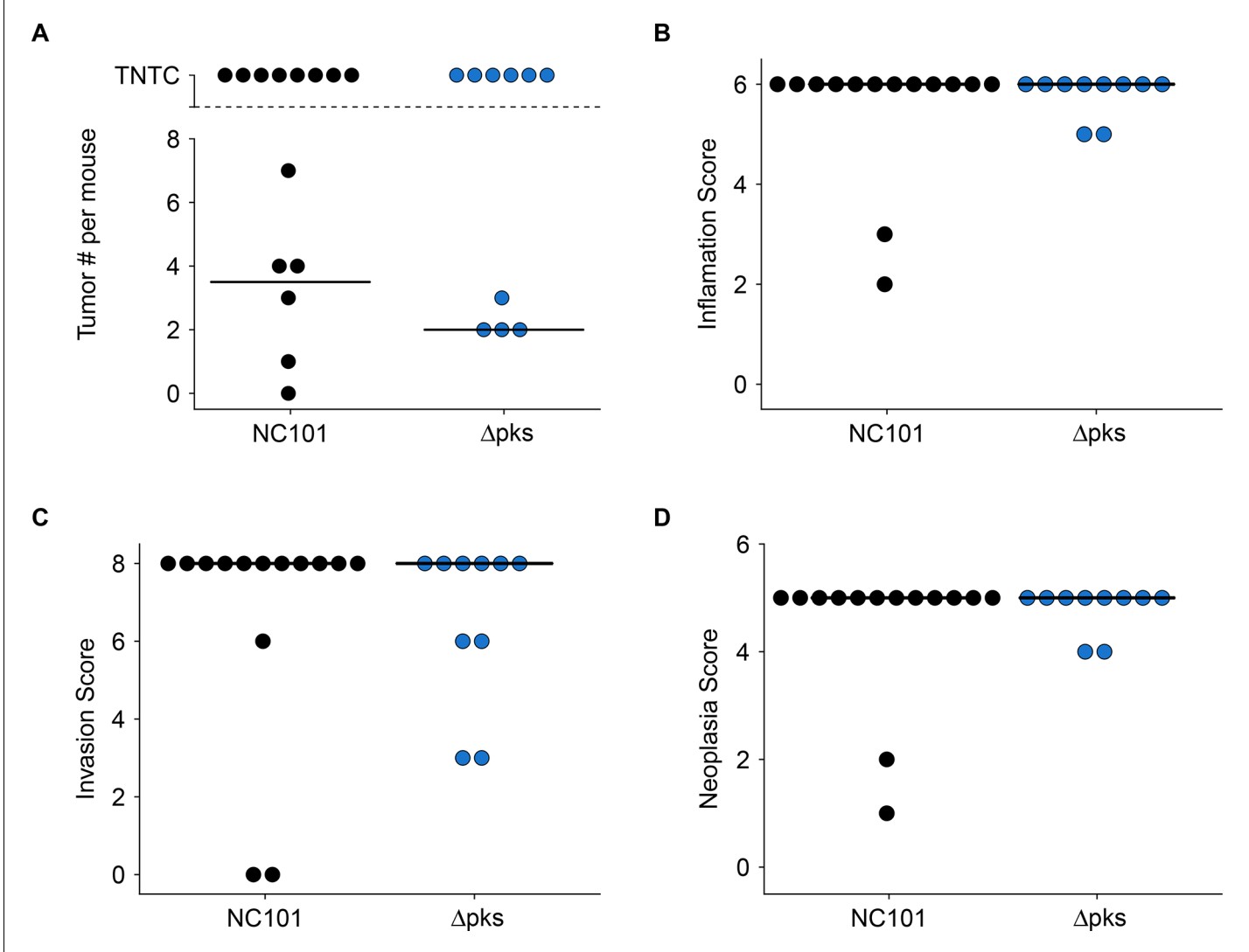

**Figure 3.** Impact of *pks* island deletion on colonic inflammation and tumorigenesis. Female and male *Il10^-/-* germ-free mice mono-associated with either *E. coli* NC101 or *E. coli* NC101Δ *pks* and treated with AOM were blindly assessed for inflammation and tumorigenesis at sacrifice. This is from the same experiment as in Figure 2. Results presented for mice that survived 14 weeks post-AOM treatment or later (Number of mice analyzed: n=14 for NC101, n=10 for NC101Δ *pks*). Dot plots where each symbol represents data from one mouse with medians reported as crossbars. One mouse inoculated with NC101Δ *pks* was found dead (186 days post-AOM) and was too autolyzed for interpretation, and thus was not included in plots. (A) Macroscopic tumor number where individual tumors were detectable. TNTC (too numerous to count) indicates mice where individual tumors could not be enumerated because of the coalescing nature of the lesions. (B) Histological inflammation scores. Exploratory analysis: Wilcoxon-Mann-Whitney test; $U = 72$, $p = 0.892$; Cliff's $d = 0.029$, 95% CI [-0.28, 0.33]. (C) Histological invasion scores. Exploratory analysis: Wilcoxon-Mann-Whitney test; $U = 80$, $p = 0.488$; Cliff's $d = 0.14$, 95% CI [-0.25, 0.49]. (D) Histological neoplasia scores. Exploratory analysis: Wilcoxon-Mann-Whitney test; $U = 72$, $p = 0.892$; Cliff's $d = 0.029$, 95% CI [-0.28, 0.33]. Additional details for this experiment can be found at https://osf.io/pm5xa/.
DOI: https://doi.org/10.7554/eLife.34364.008

The following figure supplements are available for figure 3:

**Figure supplement 1.** Gross appearance of mouse colon.
DOI: https://doi.org/10.7554/eLife.34364.009

**Figure supplement 2.** Illustrative photomicrographs of scoring system used to semi-quantify colon inflammation.
DOI: https://doi.org/10.7554/eLife.34364.010

**Figure supplement 3.** Illustrative photomicrographs of scoring system used to quantify proliferative, dysplastic, and neoplastic lesions.
DOI: https://doi.org/10.7554/eLife.34364.011

completeness and clarity in publication of experimental methodology, including experimental timing, to facilitate reproducibility between studies.

## Meta-analyses of original and replicated effects

We performed a meta-analysis using a random-effects model, where possible, to combine each of the effects described above as pre-specified in the confirmatory analysis plan (*Eaton et al., 2015*). We excluded the comparisons of inflammation, invasion, and neoplasia scores since the experimental timing between the original study and this replication attempt were not the same, preventing a direct comparison of results. To provide a standardized measure of the effect calculated for survival, a common effect size was calculated for each effect from the original and replication studies. The hazard ratio (HR) is the ratio of the probability of a particular event, in this case death, in one group compared to the probability in another group. The estimate of the effect size of one study, as well as the associated uncertainty (i.e. confidence interval), compared to the effect size of the other study provides another approach to compare the original and replication results (*Errington et al., 2014*; *Valentine et al., 2011*). Importantly, the width of the confidence interval (CI) for each study is a reflection of not only the confidence level (e.g. 95%), but also variability of the sample (e.g. *SD*) and sample size.

A meta-analysis of the intrinsic doubling times of the NC101 and NC101Δ *pks* strains was not conducted since the original study reported a single growth curve for both strains. Comparing the original and replication results, the original value reported in *Arthur et al. (2012)* for NC101 fell outside the 95% CI of the values generated during this replication attempt, while the original value for NC101Δ *pks* was within the 95% CI (*Figure 1*).

The comparison of the overall survival distributions between NC101 mono-associated mice compared to those that were mono-associated with NC101Δ *pks* resulted in a HR of 1.95, 95% CI [1.10, 3.45] for this replication attempt compared to a HR of 1.69, 95% CI [0.31, 9.09] for the original study (*Arthur et al., 2012*). Importantly, the calculation of the HR for both studies used data during the same timeframe (i.e. 18 weeks from mono-association). Both results are consistent when considering the direction of the effect, that death occurred more often in mice mono-associated with NC101Δ *pks* compared to NC101, with both effect size point estimates falling within the confidence interval of the other study. A meta-analysis (*Figure 4*) of these effects resulted in a HR of 1.92, 95% CI [1.11, 3.30], which was statistically significant ($p = 0.0188$) and implies the null hypothesis that the survival distributions for the two cohorts are the same, can be rejected.

This direct replication provides an opportunity to understand the present evidence of these effects. Any known differences, including reagents and protocol differences, were identified prior to conducting the experimental work and described in the Registered Report (*Eaton et al., 2015*). However, this is limited to what was obtainable from the original paper and through communication with the original authors, which means there might be particular features of the original experimental protocol that could be critical, but unidentified. So while some aspects, such as bacteria strain, mouse strain, and AOM dose were maintained, one aspect, experimental timing, was revealed during peer review of this Replication Study to be incorrect due to the methodology not being clearly stated in the original study, which hindered efforts to reproduce the original methodology. Thus, this replication attempt illustrates the need for methodology to be reported in sufficient detail to allow published research to be accurately compared, reproduced, and interpreted (*Glasziou et al., 2014*). Furthermore, other factors were unknown or not easily controlled for. These include variables such as mouse sex (*Clayton and Collins, 2014*), genetic heterogeneity of mouse inbred strains (*Casellas, 2011*), housing temperature in mouse facilities (*Kokolus et al., 2013*), differing compound potency and purity resulting from different stock solutions (*Davis et al., 2012*; *Kannt and Wieland, 2016*; *Neufert et al., 2007*), and genetic differences in the bacterial strains (*Kuo et al., 2009*). Environmental differences such as husbandry staff, bedding type and source, light levels, and other intangibles, all of which, by necessity, differed between the studies also affect experimental outcomes with mice (*Howard, 2002*; *Jensen and Ritskes-Hoitinga, 2007*; *Nevalainen, 2014*; *Sorge et al., 2014*). Additionally, in this replication attempt, mice were housed in isocages rather than in bubble isolators. While the difference in caging did not affect the gnotobiotic status of the mice, subtle differences in housing could result in different outcomes. Differences in pathologist's interpretation in quantification of histologic lesions is another source of variability between studies, necessitating clear delineation of criteria and terminology used for diagnosis, preferably by

illustrative photomicrographs (*Elmore et al., 2017*; *Ward et al., 2017*). Whether these or other factors influence the outcomes of this study is open to hypothesizing and further investigation, which is facilitated by direct replications and transparent reporting.

# Materials and methods

## Key resources table

| Reagent type (species) or resource | Designation | Source or reference | Identifiers | Additional information |
|---|---|---|---|---|
| Strain, strain background (Escherichia coli, NC101) | NC101 | doi:10.1126/science.1224820 | | |
| Strain, strain background (E. coli, NC101Δpks) | NC101Δpks | doi:10.1126/science.1224820 | | |
| Strain, strain background (*Mus musculus*, 129/SvEv, *Il10*$^{-/-}$) | Germ-free *Il10*$^{-/-}$ | doi:10.1126/science.1224820 | | Germ-free mice |
| Commercial assay or kit | MoBio PowerMag Microbial DNA Isolation Kit | Qiagen | cat# 27200–4 | |
| Commercial assay or kit | NEBNext Ultra DNA Library Prep Kit for Illumina | New England BioLabs | cat# E7370 | |
| Chemical compound, drug | AOM | Sigma-Aldrich | cat# A5486 | lot# SLBN5975V |
| Software, algorithm | FastQC | http://www.bioinformatics.babraham.ac.uk/projects/fastqc | RRID:SCR_014583 | version 0.11.5 |
| Software, algorithm | Trimmomatic | doi:10.1093/bioinformatics/btu170 | RRID:SCR_011848 | version 0.36 |
| Software, algorithm | SPAdes | doi:10.1089/cmb.2012.0021 | RRID:SCR_000131 | version 3.5.0 |
| Software, algorithm | ABACAS | doi:10.1093/bioinformatics/btp347 | RRID:SCR_015852 | version 1.3.1 |
| Software, algorithm | Prokka | doi:10.1093/bioinformatics/btu153 | RRID:SCR_014732 | version 1.11 |
| Software, algorithm | short-read Burrows-Wheeler Aligner | doi:10.1093/bioinformatics/btp324 | RRID:SCR_015853 | version 0.7.13 |
| Software, algorithm | Picard | http://broadinstitute.github.io/picard | RRID:SCR_006525 | version 1.130 |
| Software, algorithm | SAMtools and BCFtools | doi:10.1093/bioinformatics/btr509 | RRID:SCR_005227 | version 1.2 |
| Software, algorithm | GATK's VarieantFiltration | doi:10.1002/0471250953.bi1110s43 | RRID:SCR_001876 | version 3.3.0 |
| Software, algorithm | Artemis Comparison Tool | doi:10.1093/bioinformatics/bti553 | RRID:SCR_004507 | version 13.0.0 |
| Software, algorithm | R Project for statistical computing | https://www.r-project.org | RRID:SCR_001905 | version 3.4.4 |

As described in the Registered Report (*Eaton et al., 2015*), we attempted a replication of the experiments reported in *Figure 4A–F*, and Supplemental Figure 7 and 10 of *Arthur et al. (2012)*. A detailed description of all protocols can be found in the Registered Report (*Eaton et al., 2015*) and are described below with additional information not listed in the Registered Report, but needed during experimentation.

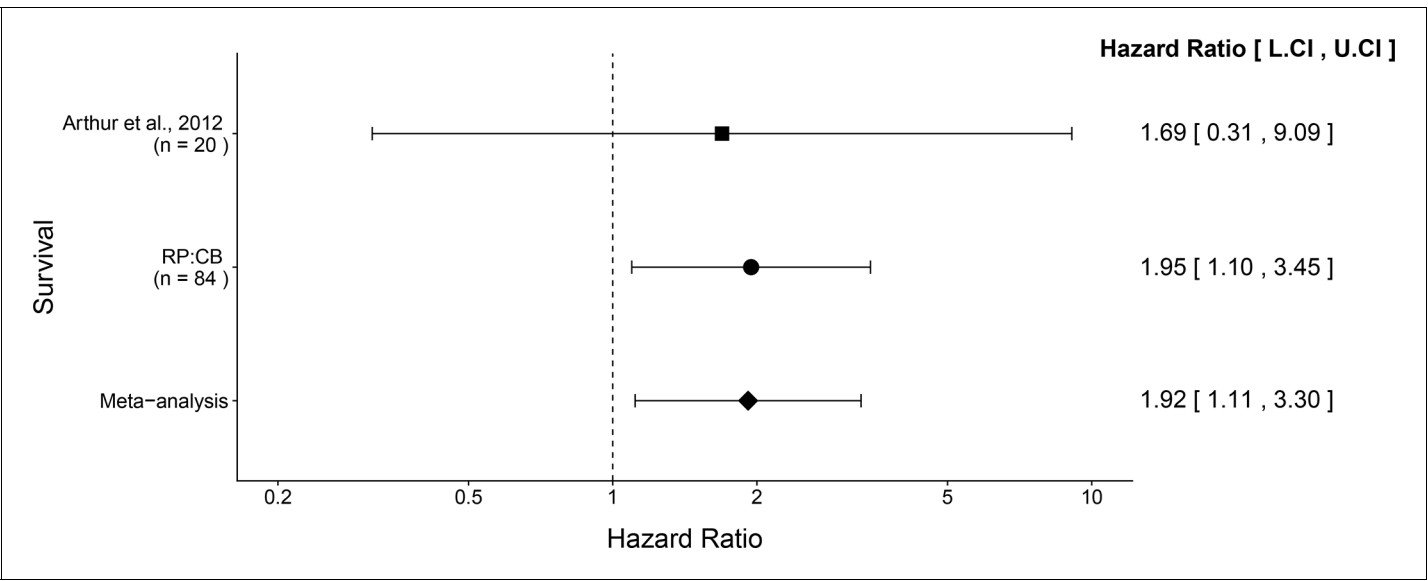

**Figure 4.** Meta-analysis of survival. Effect size and 95% confidence interval are presented for *Arthur et al. (2012)*, this replication attempt (RP:CB), and a random effects meta-analysis of those two effects. To directly compare and combine the results of both studies, the survival data during the same timeframe was used (i.e. 18 weeks from mono-association). HR greater than 1 indicates death occurred more often in NC101Δ *pks* compared to NC101, while HR less than 1 indicates the reverse. Sample sizes used in *Arthur et al. (2012)* and this replication attempt are reported under the study name. Random effects meta-analysis of HR for NC101 mono-associated mice compared to NC101Δ *pks* mono-associated mice (meta-analysis *p* = 0.0188). Additional details for this meta-analysis can be found at https://osf.io/2raud/.
DOI: https://doi.org/10.7554/eLife.34364.012

## Bacterial strains and growth conditions

*E. coli.* NC101, isolated from the feces of an inbred 129S6/SvEv background mouse raised in SPF conditions (Kim et al., 2005), and NC101Δ *pks* (*Arthur et al., 2012*), were shared by the Arthur lab, (University of North Carolina at Chapel Hill). Bacteria from an overnight culture were washed and diluted to approximately $10^8$ CFU prior to inoculation into Luria-Bertani (LB) broth. Bacteria were grown at 37°C in room air with shaking for all experiments.

## In vitro bacterial growth assay

*E. coli* strains that were grown overnight (12–16 hr) were used to inoculate 10 ml of LB broth at 1:5, 1:50, 1:500, and 1:5000 dilutions. Starting at the time of inoculation, cultures were plated in a 96 well plate in technical triplicate and were measured every 30 min at 600 nm absorbance using a microplate spectrophotometer. Plates were maintained at 37°C during the assay. LB broth was used to determine the background, which was subtracted from the readings. Technical repeats were averaged for each biological repeat. To summarize the growth characteristics (e.g. doubling time), values for each biological repeat were fit to the standard form of the logistic equation common in ecology and evolution using the *Growthcurver* R package (*Sprouffske and Wagner, 2016*) and R software (RRID:SCR_001905), version 3.4.4 (*Core Team, 2018*).

## PCR detection of *pks* island

Bacterial DNA was isolated from overnight cultures using a DNeasy Blood and Tissue kit according to manufacturer's instructions (Qiagen, cat# 69504). Fecal material was processed for DNA extraction by resuspending pellets in lysis buffer supplemented with 20 mg/ml lysozyme, incubated at 37°C for 30 min and then supplemented with 10% SDS and 350 μg/ml proteinase K. DNA was extracted with a DNeasy Blood and Tissue kit according to the manufacturer's instructions. DNA was quantified using a NanoDrop spectrophotometer (Thermo Fisher Scientific, cat# 2000C). PCR was performed on a MJ Mini Gradient Thermal Cycler (BioRad, cat# PCT-1148) and Opticon Monitor software (RRID:SCR_014241). PCR reactions were performed using primers specific for the 5' and 3' end of the *pks* island, colibactin, and 16S rRNA, with sequences listed in the Registered Report

(*Eaton et al., 2015*). Reaction volumes were 50 μl and consisted of 5 μl 10X Taq polymerase Master Mix supplemented with 1.5 mM $Mg^{2+}$ and 0.5 μl Taq polymerase (Sigma-Aldrich, cat# D9307), 0.05 mM dNTPs, 0.05 μM forward and reverse primers, and 2 μl DNA diluted in water. A negative control without DNA was also included. PCR cycling conditions were: 1 cycle 95°C for 5 min – 35 cycles (or 27 cycles) 95°C for 45 s, 56°C 45 s, 72°C 45 s – 1 cycle 72°C for 10 min. PCR reactions were run on a 1.5% agarose gel to visualize if a product of expected size was produced.

## Genome sequencing data processing and assembly

Bacterial DNA was extracted with the MoBio PowerMag Microbial DNA Isolation Kit (Qiagen, cat# 27200-4) according to the manufacturer's instructions and prepared for sequencing on an Illumina MiSeq instrument (San Diego, California) (MiSeq run parameters can be found at https://osf.io/fnu62/) using the NEBNext Ultra DNA Library Prep Kit for Illumina (New England BioLabs, cat# E7370) and sample-specific barcoding. Library preparation and sequencing were performed at the Center for Microbial Systems at the University of Michigan. Samples were evaluated for contamination and excessive low quality sequence with FastQC (RRID:SCR_014583), version 0.11.5 (*Andrews, 2016*), and processed using Trimmomatic (RRID:SCR_011848), version 0.36 (*Bolger et al., 2014*), to trim low quality bases and remove reads with poor average quality scores. *De novo* genome assemblies were generated for each sample by running SPAdes (RRID:SCR_000131), version 3.5.0 (*Bankevich et al., 2012*), on trimmed sequencing reads. For comparison, the assembly for NC101Δ *pks* was ordered relative to NC101 using ABACAS (RRID:SCR_015852), version 1.3.1 (*Assefa et al., 2009*), and base genome annotation was assigned with Prokka (RRID:SCR_014732), version 1.11 (*Seemann, 2014*).

## Variant detection

Variants were identified by: (1) mapping filtered reads from NC101Δ *pks* to the assembled NC101 reference genome (GenBank Accession number: AM229678.1) using the short-read Burrows-Wheeler Aligner (BWA) (RRID:SCR_015853), version 0.7.13 (*Li and Durbin, 2009*), (2) discarding PCR duplicates with Picard (RRID:SCR_006525), version 1.130 (http://broadinstitute.github.io/picard), and (3) calling variants with SAMtools and BCFtools (RRID:SCR_005227), version 1.2 (*Li, 2011*). Variants were filtered from raw results using GATK's (RRID:SCR_001876) VariantFiltration, version 3.3.0 (QUAL > 100, MQ > 50, >10 reads supporting variant, FQ < 0.025) (*Van der Auwera et al., 2013*). In addition, a custom python script (https://osf.io/jgqdb/) was used to filter out single nucleotide variants that were: (1) < 5 bp in proximity to indels or (2) < 10 bp in proximity to another variant.

## Large indel detection

To identify genomic regions differing between NC101 and NC101Δ *pks*, bi-directional BLAST queries were performed between the contigs in the genome assemblies. Regions found to be unique to either genome were verified by mapping reads using BWA and visually verifying that no reads map to putative unique genomic regions using the Artemis Comparison Tool (RRID:SCR_004507), version 13.0.0 (*Carver et al., 2005*).

## AOM/*Il10*^-/- animal model

All animal procedures were approved by the Michigan University IACUC# 7291 and were in accordance with Michigan University's policies on the care, welfare, and treatment of laboratory animals. Blinding occurred during histopathology scoring of inflammation and tumorigenesis scoring. Mice were randomized for mono-association.

Germfree (GF) *Il10*^-/- mice of the 129S6/SvEV background were originally from the National Gnotobiotic Rodent Resource Center at the University of North Carolina, Chapel Hill and shipped in germ free shipping containers (Taconic) to the Germ-Free & Gnotobiotic Mouse Facilities at Michigan University. Mice for this study were born and raised in GF isolators until they reached the age of 7-12 weeks. GF status was verified by bacterial culture, Gram stain, mold trap, and 16S bacterial PCR. Gram stain and bacterial culture were performed at every isolator entry. Mice were aseptically removed from the isolators, randomly assigned to be mono-associated with NC101 or NC101Δ *pks*, and housed in sterile isocages (Tecniplast), where they stayed throughout the study. After the mice were moved to the isocages, gnotobiotic status was verified weekly by Gram stain and bacterial

culture, while mold traps were monitored daily to confirm GF status. Early in the study a few iso-cages were contaminated and had to be removed from the study. This did not affect the other iso-cages which remained gnotobiotic.

A similar distribution of male and female mice were assigned to both strains (NC101: female=21, male=18; NC101Δ *pks*: female=23, male=22). Mice of the same sex and mono-associated with the same bacteria strain were caged together (2-4 mice per isocage). Mice were mono-associated by oral gavage and rectal swabbing with 200 µl of an overnight log phase bacterial culture at a concentration of $2x10^9$ colony forming units (CFU)/ml. Gnotobiotic status was verified weekly by Gram stain and culture of fecal contents. For culture, sterile swabs were used to transfer fecal material to sheep blood agar plates, which were incubated at 37°C under aerobic or anaerobic conditions. For Gram staining, swabs with fecal material were transferred to glass slides and spread evenly in a thin layer. Slides were air-dried, heat-fixed, and Gram stained using a BBL Gram Stain Kit according to manufacturer's instructions (BD Biosciences, cat# BD 212539). Throughout the experiment, mice were observed at least once daily and weighed weekly. Moribund and dead animals were necropsied unless autolysis precluded any interpretation. Mice were offered Purina Lab Diet 3500 (the same diet that was used in the original study) and sterile water *ad libitum*. They were housed on Tek-Fresh bedding (Envigo) and offered Enviro-dri nesting material (Shepherd Specialty Papers) as enrichment.

Four weeks after mono-association, colonization and bacterial strain were verified by culture and PCR. At the same time, weekly intraperitoneal injections of 10 mg/kg AOM diluted to a final dilution of 2.5 mg/ml (Sigma-Aldrich, cat# A5486, lot# SLBN5975V) were initiated and continued until mice received six injections. The same lot of AOM was used for the entire study with 25 mg/ml aliquots stored at −80°C until use. One vial was used for each injection day and any remainder discarded to avoid unnecessary freeze-thaw cycles. Mice were euthanized and necropsied 18 weeks after the sixth AOM injection as prespecified in the Registered Report (*Eaton et al., 2015*), or when they became moribund. At necropsy, gross lesions were recorded and photographed, if present, and as far as possible, samples were collected for culture and PCR of cecal contents, and for histopathologic evaluation of colon lesions. Colon lesions were scored for all mice that survived to 14 weeks or more after the last AOM injection. The experimental timeline is illustrated in *Figure 2A*.

## Determination of *E. coli* CFU

To quantify cultures for mono-association, bacteria were cultured overnight (12–16 hr) at 37°C in LB broth to log phase growth and CFU/ml was estimated based on $OD_{600}$ readings. Cultures were adjusted to the desired density on the OD reading and mice were mono-associated. For precise determination of CFU/ml, an aliquot of the culture was quantified by serial dilution (10 fold dilutions in LB broth). 100 µl of each dilution were plated on LB agar and incubated at 37°C for 24 hr under aerobic conditions. On plates with discrete colonies, the number of colonies were counted and results were expressed as CFU/ml of contents.

To quantify colonization of mice, samples of feces were aseptically collected into pre-weighed, sterile tubes (average weight of feces was 0.05 g). Samples were resuspended as a slurry in 1 ml sterile LB broth and serially diluted (10 fold dilutions in LB broth). 100 µl of each dilution were plated on LB agar and incubated at 37°C for 24 hr under aerobic conditions. On plates with discrete colonies, the number of colonies were counted and results were expressed as CFU/ml of contents.

## Histopathology

Mice were sacrificed at the indicated time points and colon (proximal, mid-proximal (transverse), distal) and cecum, liver, spleen, and any gross lesions were collected for histological evaluation. Colons were blindly examined macroscopically for tumors by a board-certified veterinary pathologist. Tissues were fixed in 10% neutral buffered formalin for 24–48 hr, with colon tissue Swiss-rolled from the proximal to the distal end. The fixed tissue was embedded in paraffin, sectioned at five microns, and stained with hematoxylin and eosin (H&E) as described in the Registered Report (*Eaton et al., 2015*). Individual sections were blindly examined microscopically (Olympus BX41) by a board-certified veterinary pathologist. Colon sections were scored for inflammation (*Rath et al., 1996*) and invasion using the same scoring criteria as the original study (*Arthur et al., 2012*) and specified in the Registered Report (*Eaton et al., 2015*). Histopathology scoring, images, and additional protocol details are available at https://osf.io/pm5xa/.

Scoring of inflammation was based on a study by *Rath et al. (1996)* where inflammation was semi-quantified. Criteria are summarized as follows: score 1 = increased inflammation in the lamina propria, decreased goblet cells and mucosal thickening (all mild); score 2 = moderately increased inflammation, decreased goblet cells and mucosal thickening with the addition of mild submucosal inflammation: score 3 = severely increased inflammation, decreased goblet cells and mucosal thickening with moderate submucosal inflammation and mild destruction of architecture; score 4 = severely increased inflammation, decreased goblet cells and mucosal thickening, and moderate destruction of architecture, score 4.5–6 was used based on the presence of ulcers and crypt abscesses. For the current study, increased mucosal thickening was interpreted to mean mucosal hypertrophy, and destruction of architecture was interpreted to mean atrophy or loss of epithelial cells or glands, fibrosis, or collapse of lamina propria. The entire length of the colon was examined and an overall score assigned. The scoring system for inflammation is illustrated in *Figure 3—figure supplement 2*.

Dysplasia was scored as described in the original study with minor clarifications. Low and high-grade dysplasia, intra-epithelial neoplasia, adenoma, herniation, invasion, and adenocarcinoma, not defined in the original publication, were here defined based on published criteria (*Boivin et al., 2003*; *Washington et al., 2013*). Invasion was distinguished from herniation based on the level of scirrhous response and cellular atypia (*Boivin et al., 2003*; *Washington et al., 2013*). Altered crypt foci, a gross characteristic, was not evaluated in this study. Gastrointestinal intraepithelial neoplasia was interpreted to mean carcinoma *in situ*, also referred to as small non-invasive adenomas or individually transformed crypts (*Washington et al., 2013*). The scoring system for neoplasia and invasion is illustrated in *Figure 3—figure supplement 3*. Neoplasia was scored taking into account the entire colon section and not simply the most severe lesion, and summarized as follows: 0 = no dysplasia; 1 = mild dysplasia characterized as aberrant crypt foci, +0.5 for multiples; 2 = moderate dysplasia characterized as gastrointestinal neoplasia, +0.5 for multiples; 3 = severe or high grade dysplasia characterized as adenoma, restricted to the mucosa; 4 = invasive adenocarcinoma, invading into or through the muscularis mucosa; and 5 = fully invasive adenocarcinoma, full invasion through the submucosa and into or through the muscularis propria (*Arthur et al., 2014*).

## Statistical analysis

Statistical analysis was performed with R software (RRID:SCR_001905), version 3.4.4 (*Core Team, 2018*). All data, csv files, and analysis scripts are available on the OSF (https://osf.io/y4tvd/). Confirmatory statistical analysis was pre-registered (https://osf.io/yt9ki/) before the experimental work began as outlined in the Registered Report (*Eaton et al., 2015*) with any other analysis indicated as exploratory. Data were checked to ensure assumptions of statistical tests were met. The nonparametric Wilcoxon-Mann-Whitney test was used for the inflammation and tumorigenesis scoring analysis because ordinal scoring data do not meet the assumption of a normal distribution (*Gibson-Corley et al., 2013*). That is, while a number is used for the scoring it represents non-numeric concepts like 'severe or high grade dysplasia characterized as adenoma, restricted to the mucosa'. Thus, non-parametric approaches are the best way to describe results from these data, especially with small sample sizes (*Baker et al., 2014*). The asymmetric confidence intervals for the overall Cliff's *d* estimate was determined using the normal deviate corresponding to the (1-alpha/2)[th] percentile of the normal distribution (*Cliff, 1993*). A meta-analysis of a common original and replication effect size was performed with a random effects model and the metafor R package (*Viechtbauer, 2010*) (https://osf.io/2raud/). The original study data presented in *Figure 4A–E* was extracted *a priori* from the published Figure by estimating the value of each symbol based on the scoring criteria described in the original study methods and shared by the original authors. The data were published in the Registered Report (*Eaton et al., 2015*) and used in the power calculations to determine the sample size for this study. To provide a comparison of the replication results to the original study for in vitro bacterial growth and animal survival, the values reported in the original study in Supplemental Figure 7 and 10 were estimated.

## Data availability

Additional detailed experimental notes, data, and analysis are available on OSF (RRID:SCR_003238) (https://osf.io/y4tvd/; *Eaton et al., 2018*). This includes the R Markdown file (https://osf.io/ektn3/)

that was used to compose this manuscript, which is a reproducible document linking the results in the article directly to the data and code that produced them (*Hartgerink, 2017*). The Whole Genome sequencing data generated during this study has been deposited at NCBI SRA under the Bioproject accession PRJNA481682. The genome assemblies have been deposited at Genbank under the accession QVAD00000000 and QVAE00000000.

## Deviations from registered report

The in vitro bacterial growth assay was performed at multiple dilutions in addition to the 1:500 dilution specified in the Registered Report. This was done to test if there was an impact of the starting density on growth kinetics. The number of mice enrolled in the study was increased from an estimated total of 30 to 84. This was largely due to early deaths, during AOM treatment and an attempt to obtain the prespecified number of mice 18 weeks after the last AOM injection. Scoring of colon sections for neoplasia was performed using a slight variation of the scale listed in the Registered Report and methods of the original study (*Arthur et al., 2012*) to ensure a direct comparison was made between the original study and this replication. The values assigned to 'adenocarcinoma, invasion through the muscularis mucosa' were changed from 3.5 to 4, while the values assigned to 'adenocarcinoma, full invasion through the submucosa and into or through the muscularis propria' were changed from 4 to 5. These changes justify the reported scores of 5 in *Figure 4B* of *Arthur et al. (2012)*, which would not be possible unless the scale was changed, and was the scale described in a more recent paper by the original authors (*Arthur et al., 2014*). The statistical analyses proposed in the Registered Report for the scoring and survival data were not able to be performed due to differences in experimental timing that we were informed about during peer review of this Replication Study manuscript, but were not revealed during informal review and feedback by the authors of the original paper during experimental planning or during peer review of the Registered Report. The exploratory analyses for survival took into account the entire study period (mono-association to 18 weeks after the last AOM treatment) and the same period as the original study (18 weeks from mono-association). The latter analysis was used in the meta-analysis of the two studies. The exploratory analyses for the scoring data were performed using nonparametric tests as described above. This differs from the original study that used parametric tests to analyze non-parametric scoring data (i.e. ordinal data). Since we observed a higher death rate for mice mono-associated with NC101Δ *pks* compared to NC101, we performed whole genome sequencing on the two *E. coli* strains to examine if any genetic differences existed beyond the deletion of the *pks* island. Additional materials and instrumentation not listed in the Registered Report, but needed during experimentation are also listed.

## Acknowledgements

The Reproducibility Project: Cancer Biology would like to thank Dr. Janelle C Arthur (University of North Carolina at Chapel Hill), for sharing critical information, data, and reagents, specifically the *E. coli* NC101 and NC101Δ *pks* strains, as well as the *Il10*[-/-] germ-free mice (grants: 5-P39-DK034987 and 5-P40-OD010995). We would like to thank Clinton Fontaine and Nicholas Pudlo for performing some of the *in vitro* work and Sara Poe, Chriss Vowles, Trisha Denike, and Natalie Anderson who performed the animal work. We would also like to thank the following companies for generously donating reagents to the Reproducibility Project: Cancer Biology; American Type and Tissue Collection (ATCC), Applied Biological Materials, BioLegend, Charles River Laboratories, Corning Incorporated, DDC Medical, EMD Millipore, Harlan Laboratories, LI-COR Biosciences, Mirus Bio, Novus Biologicals, Sigma-Aldrich, and System Biosciences (SBI).

## Additional information

### Group author details

**Reproducibility Project: Cancer Biology**
**Elizabeth Iorns**: Science Exchange, Palo Alto, United States; **Rachel Tsui**: Science Exchange, Palo Alto, United States; **Alexandria Denis**: Center for Open Science, Charlottesville, United States;

Nicole Perfito: Science Exchange, Palo Alto, United States; Timothy M Errington: Center for Open Science, Charlottesville, United States; Elizabeth Iorns: Science Exchange, Palo Alto, United States; Rachel Tsui: Science Exchange, Palo Alto, United States; Alexandria Denis: Center for Open Science, Charlottesville, United States; Nicole Perfito: Science Exchange, Palo Alto, United States; Timothy M Errington: Center for Open Science, Charlottesville, United States

## Competing interests

Kathryn Eaton: Germ-Free and Gnotobiotic Mouse Facilities, University of Michigan Medical School was a Science Exchange associated lab. Reproducibility Project: Cancer Biology: EI, RT, NP: Employed by and hold shares in Science Exchange Inc. The other authors declare that no competing interests exist.

## Funding

| Funder | Author |
| --- | --- |
| Laura and John Arnold Foundation | Reproducibility Project: Cancer Biology |

The funders had no role in study design, data collection and interpretation, or the decision to submit the work for publication.

## Author contributions

Kathryn Eaton, Ali Pirani, Evan S Snitkin, Acquisition of data, Analysis and interpretation of data, Drafting or revising the article; Reproducibility Project: Cancer Biology, Analysis and interpretation of data, Drafting or revising the article

## Author ORCIDs

Alexandria Denis http://orcid.org/0000-0002-1210-2309
Timothy M Errington https://orcid.org/0000-0002-4959-5143
Alexandria Denis http://orcid.org/0000-0002-1210-2309
Timothy M Errington https://orcid.org/0000-0002-4959-5143

## Ethics

Animal experimentation: All animal procedures were approved by the Michigan University IACUC# 7291 and were in accordance with Michigan University's policies on the care, welfare, and treatment of laboratory animals.

## Decision letter and Author response

Decision letter https://doi.org/10.7554/eLife.34364.025
Author response https://doi.org/10.7554/eLife.34364.026

# Additional files

## Supplementary files

• Transparent reporting form
DOI: https://doi.org/10.7554/eLife.34364.013

• Reporting standard 1
DOI: https://doi.org/10.7554/eLife.34364.014

## Data availability

Additional detailed experimental notes, data, and analysis are available on OSF (RRID:SCR_003238) (https://osf.io/y4tvd/; Eaton et al., 2018). This includes the R Markdown file (https://osf.io/ektn3/) that was used to compose this manuscript, which is a reproducible document linking the results in the article directly to the data and code that produced them (Hartgerink, 2017). The Whole Genome sequencing data generated during this study has been deposited at NCBI SRA under the Bioproject

accession PRJNA481682. The genome assemblies has been deposited at Genbank under the accession QVAD00000000 and QVAE00000000.

The following datasets were generated:

| Author(s) | Year | Dataset title | Dataset URL | Database, license, and accessibility information |
|---|---|---|---|---|
| Eaton K, Pirani A, Snitkin ES, Iorns E, Tsui R, Denis A, Perfito N, Errington TM | 2018 | Replication Study: Intestinal inflammation targets cancer-inducing activity of the microbiota | https://www.ncbi.nlm.nih.gov/bioproject/481682 | Publicly available at NCBI BioProject (accession no. PRJNA481682) |
| Eaton K, Pirani A, Snitkin ES, Iorns E, Tsui R, Denis A, Perfito N, Errington TM | 2018 | Escherichia coli strain NC101-PKS, whole genome shotgun sequencing project | https://www.ncbi.nlm.nih.gov/nuccore/QVAD00000000 | Publicly available at NCBI Nucleotide (accession no. QVAD00000000) |
| Eaton K, Pirani A, Snitkin ES, Iorns E, Tsui R, Denis A, Perfito N, Errington TM | 2018 | Escherichia coli strain NC101-WT, whole genome shotgun sequencing project | https://www.ncbi.nlm.nih.gov/nuccore/QVAE00000000 | Publicly available at NCBI Nucleotide (accession no. QVAE00000000) |
| Eaton K, Pirani A, Snitkin ES, Iorns E, Tsui R, Denis A, Perfito N, Errington TM | 2018 | Study 41: Replication of Arthur et al., 2012 (Science). | http://dx.doi.org/10.17605/OSF.IO/Y4TVD | Available at OSF under a CC Attribution 4.0 International Public License |

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
