## [Decision Letter]

Thank you for submitting your article "Replication Study: Intestinal inflammation targets cancer-inducing activity of the microbiota" for consideration by *eLife*. Your article has been reviewed by Wendy Garrett as the Senior Editor, a Reviewing Editor, and three reviewers. The reviewers have opted to remain anonymous.

The reviewers have discussed the reviews with one another and the Reviewing Editor has drafted this decision to help you prepare a revised submission.

Summary:

In this replication work, several features of the original publication by Arthur et al. were replicated including similar mouse colonization with PKS+ or ΔPKS NC101 *E. coli*; no effect of the ΔPKS on bacterial growth; and strain confirmation including PKS island deletion. However, for the critical comparisons of inflammation, invasion and neoplasia, the replication attempt did not model the kinetics of the mouse model as reported by Arthur et al. This appeared to be due to at least two technical features including increased murine death likely associated with the AOM batch used in the experiments, a known but not highly visible understanding from the literature and use of an experimental evaluation time course differing from the Arthur et al., paper. Further, contamination of isolators occurred that further limited the data interpretation. Thus, as conducted, the study did not achieve replication of the conditions of the paper of Arthur et al. As presented by the reviewer comments, the paper discussion and conclusion should be revised to focus on critical technical and analytical issues that emerged and must be considered by other investigators if working with this model and these bacterial strains.

Essential revisions:

As pointed out by all reviewers, the attempt at replication primarily revealed the technical aspects that must be controlled for in working with this mouse model in future experiments. A careful standardization of the protocol/disease course is required in individual laboratories before comparing different *E. coli* strains. Key features of the model to be verified before strain testing include low mortality rates, distinguishable individual tumors at the endpoint and additional controls including non-colonized AOM-injected-only control mice and mono-colonization-only and possibly AOM-untreated control mice. Revising the paper to focus on the technical limitations rather than that the replication attempt led to results differing from the original paper seems prudent. Ultimately, since the biology and kinetics of the Arthur et al., model were not faithfully replicated, these data are unable to address whether, under the conditions of the original paper, NC101 and ΔPKS NC101 differ in inflammation, invasion and neoplasia. This final point should be made clear to the readers of the paper.

*Reviewer #1:*

Eaton et al., try to reproduce key findings from Arthur et al., 2012 paper.

All of the experiments proposed in the initial *eLife* Registered Report have been performed and analyzed. The observed results and conclusions according to the authors are the following:

1) Authors established colonization of germ free *Il10^-/-^* mice with the PKS+ or ΔPKS NC101 *E. coli* as in original paper.

2) AOM injection in the colonized mice induced colorectal tumorigenesis as in original paper.

3) Similarly to Arthur et al., deletion of PKS island in *E. coli* genome did not affect colonization and bacterial growth of *E. coli* in germ free mice.

4) Authors conclude that ΔPKS induces similar level of inflammation (agreement with original study) and similar level of tumorigenicity (multiplicity, invasion- in disagreement with original study) in comparison with PKS+ colonized animals.

5) Authors claim that overall inflammation and other histopathological observations were more severe in the replication attempt.

6) Overall mortality of mice throughout the experiment was much higher in the replication attempt.

7) Meta-analyses suggest that there is no difference between ΔPKS and PKS+ colonized mice, contrary to the original paper by Arthur et al., 2012.

8) Authors hint at the conclusion that the main observation of Arthur et al. paper is not reproduced (more neoplasia, tumors and invasion in PKS+ colonized mice) but also conclude that results seen in replication attempt may be confounded by the increased severity of inflammation and mortality of animals during the experiment.

Overall authors performed all of the experiments proposed. Data analysis is solid. This reviewer is not an expert biostatistician, so no professional comments on the statistics part can be made.

Specific comments:

1) Mortality seen on Figure 2A. These curves may represent two different types of mortalities (causes)- AOM induced mortality early on and then inflammation and/or tumor load induced mortality later.

Early mortality in the Replication experiment is clearly higher than in Arthur et al. (>50% vs less than 30%).

What is presented on Figure 2B may be very similar to the data on Supplementary Figure 10 by Arthur et al., (although original paper mortality curve is not well described in terms of what timepoints there are plotted), if mortality between day 0 and 10 is still attributed to the action of AOM. Then long time there is no mortality (days 20 to 80) in both Replication experiment and in Arthur et al. Then mice in Arthur et al., seemingly start to die, but they are immediately collected (all of them), and experiment stopped, so we do not know what happens to mortality. In Replication experiment past day 90 mortality starts but mice are euthanized according to symptoms or dying, so there is an impression of high mortality.

2) One important discrepancy here is that in Arthur et al., 14 wk and 18 wk point mice were collected when they still were not dying, indicating that probably there is still a room for a difference in inflammation and tumor load/invasion.

In Replication attempt, it seems from the Figure legend that mice were mostly collected and "declared dead" when they were already sick (right thing to do in terms of following IACUC regulation) but one could think that by the time PKS+ and ΔPKS mice are already equally very sick, tumor load and inflammation may be already too high and indistinguishable. Indeed, authors note that overall role of inflammation, tumor load and mortality in their experiment was higher than in initial experiment.

3) Another thing noted during initial planning of experiments and review of Registered Report is that AOM quality may vary. Indeed, different lots of AOM may have different amount of "acting" AOM and sometimes it”s too much- more toxicity/mortality and more tumors in residual mice or poor decayed AOM- more toxicity/mortality and less tumors at the end. Seemingly, in Replication report higher mortality (presumably AOM induced) and higher tumor load is seen.

4) Since AOM metabolism and potentially bacterial and liver metabolism may be involved, the diets used for germ free mice should be compared and that should be discussed

5) Using Chi- values for tumor numbers in Replication study when tumors cannot be enumerated (why?) further complicates the interpretation.

6) Another important caveat which should be acknowledged also stems from high mortality. When mortality is that high, there may be a natural selection for mice with lower inflammatory response and lower tumor load- and the mice where potentially there was (could be) a difference have been kept for too long under these conditions of inflammation and AOM lot. This is not very scientific argument, I know, but after this Replication attempt is published, the field will need all of the possible scenarios/explanations discussed.

7) Inflammation is known to promote cancer. PKS+ bacteria is proposed to be more carcinogenic than PKSΔ. However, under the conditions of stronger inflammation (observed in Replication attempt), Inflammation arguably can become more important than PKS status. PKS is genotoxic, but so are the products of excessive inflammatory responses (ROS, RNI etc.). One could argue that under stronger inflammation the presence or absence of PKS becomes less relevant as for its effect towards tumor progression. Therefore, comparison of this (Replication) and Arthur et al. studies with regard to tumor characteristics only makes sense if the same levels of inflammation are achieved.

In my view, for these experiments the technical attempt to reproduce the results has been attempted but based on the confounded data it is difficult to judge whether important conceptual results are reproduced or not (i.e. whether PKS+ is superior at tumor progression).

The only thing which can be concluded, in my view, is that during the Replication attempt, the team did not achieve the same levels of AOM toxicity/action, the same levels of inflammation and tumor load (more); and observed higher mortality. Under these circumstances it is still not clear whether PKS+ *E. coli* induces more tumors and more aggressive tumors; than ΔPKS controls.

*Reviewer #2:*

The authors of the this replication study of the paper by Artur et al., discuss in detail all experiments performed in their study based on the initial registered replication plan and also justify all changes done compared to that protocol. They have also included in their Introduction all relevant articles published since the Registered Report was published.

The major experimental findings of the study are the following:

1) In Figure 1 the authors compare in vitro the growth of *E. coli* strains NC101 and NC101𝛥pks (both obtained from the laboratory of the original study) and observe that pks deletion does not affect the growth curve, in agreement to what reported by Arthur et al. Therefore, the deletion of the pks island does not affect *E. coli* growth in vitro.

2) The authors verify that *E. coli* NC101𝛥pks bears a genetic deletion of the pks island and shows no other genetic variants or indels compared to the NC101 strain. (Figure 1—figure supplement 1).

3) The authors validate that *E. coli* strains NC101 and NC101𝛥pks display a similar growth *in vivo* in monocolonized germ-free *Il10^-/-^* mice, as assessed by CFU/ml in the feces. (Figure 1—figure supplement 1).

4) In Figure 2 the authors compare the survival curves of germ-free *Il10^-/-^* mice that are mono-associated with *E. coli* NC101 or *E. coli* NC101𝛥pks. Extensive lethality was observed in mice monocolonized with both strains, starting after the first AOM injection. No statistically significant differences were observed between mice colonize with the two strains in terms of survival over the course of the experiment.

5) In Figure 3 the authors count the number of tumors/mouse in surviving mice in which individual tumors could be distinguished and also evaluate the extent of inflammation in them.

6) In Figure 4 the authors present the results of metanalyses of their results and the ones by Arthur et al., as described in the registered replication plan.

Major points:

The major aim of the present study is to compare the potential of the *E. coli* strains NC101 and NC101𝛥pks in inducing intestinal inflammation and tumor formation in monocolonized GF *Il10^-/-^* mice in order to evaluate the reproducibility of the experiments published by Arthur et al. For this comparison to be performed the kinetics of the model should be similar to what was reported by Arthur et al. In the present study mice mono-colonized with both *E. coli* strains display a much higher lethality and an exacerbated phenotype by histopathology as compared to the report by Arthur et al. A very low number of mice met the endpoint of the study. Because of these issues no conclusions can be drawn regarding the potential of *E. coli* NC101 vs NC101𝛥pks to induce intestinal inflammation and tumor formation in monocolonized GF *Il10^-/-^* mice.

In the Abstract the authors first mention that "Mono-association […] resulted in similar levels of intestinal inflammation and tumorigenesis; whereas the original study reported decreased tumor multiplicity […]" and secondly, they mention that this replication study showed more severe histopathological observations and a much lower survival rate compared to the original study. This structure of the Abstract is misleading to the reader. The strong differences in the kinetics of the experiment compared to the original report and the lack of power of the current study to assess the reproducibility of the original findings should be mentioned first. The observations regarding tumor number and histopathological features could be discussed as indicative of a lack of difference in the context of a severely exacerbated disease course.

In my opinion, the major conclusion of this study is that the AOM/*E. coli* NC101 *Il10^-/-^* mono-colonization model is characterized by important technical limitations that should be carefully controlled in future studies. A careful standardization of the protocol/disease course is required in individual laboratories before comparing different *E. coli* strains. This should lead to low mortality rates and distinguishable individual tumors at the endpoint. Including non-colonized AOM-injected-only control mice and mono-colonization-only, AOM-untreated control mice is essential for the correct interpretation of results in this model. Such factors should be considered for the experimental design of future studies.

Additional points:

The comparison of the death rates between this study and Arthur et al. is vague in this part of the manuscript despite the fact that this is the major limitation of this replication attempt. The authors should clearly mention the percentages of mice meeting the endpoint in the two studies.

Mortality was highest during AOM treatment independently of the bacterial strain used whereas mice started dying right after the first AOM injection. The interpretation of these observations is difficult because of the absence of AOM-only controls: lethality over the injection period may be an effect of AOM toxicity alone or of AOM in combination with the effect of E.coli. It may also be an effect of the injection process itself (bleeding of internal organs for example). This is an additional technical variable that should be mentioned in the Discussion.

The authors mention that mortality during AOM injections was greater in mice colonized with NC101𝛥pks. Is this difference statistically significant? This should be mentioned and if significant further discussed.

The authors discuss sex-related differences which may confound the differences observed by Arthur et al. where 4 female and 8 male mice mono-associated with NC101 and 8 male mice mono-associated with NC101𝛥pks were analyzed. What is the result of these analyses if only male mice are considered? (n = 8 vs 8)?

The study is not powered enough to draw conclusions from the histopathological analyses because of the small number of mice resulting from an exacerbated disease course in the experiment.

Figure 2A: Log-rank test not shown in the legend.

*Reviewer #3:*

In this replication study, the authors had conducted the experiments under the conditions as close as possible to the original study; correctly reported everything they observed in the study including both the experiments that they were able and unable to replicate; and reasoned that they were unable to reproduce the intestinal inflammation and tumorigenesis results may be due to the overall disease development in this replication study was much more severe than the original study.

It appears that the authors of this replication study had tried the best to conduct the experiments under the conditions as close as possible to the original study. For instance, bacterial strains were shared by the Arthur Lab (subsection “Bacterial strains and growth conditions”), germ-free *Il10^-/-^*mice were derived from the same germ-free colony used in the original study) (subsection “Intestinal tumorigenesis and inflammation of germ-free *Il10^-/-^* mice mono-associated with E. 121 coli NC101 or NC101𝛥pks”), and the used AOM dose, 10 mg/kg, was the same as the original study (the same vendor, same catalog number, but with different lot number). AOM administration (the same vendor, same catalog number, same dosage used, but with different lot number) had resulted in a significantly higher mortality rate in this replication study compared with the original study. As cited by the authors (subsection “Intestinal tumorigenesis and inflammation of germ-free *Il10^-/-^* mice mono-associated with E. 121 coli NC101 or NC101𝛥pks”), a previous study (Bissahoyo, 2005) showed that 33 mice were treated with 10 mg/kg of AOM, and "no premature loss of mice" was observed at "six months after the first AOM dose". This appears in line with the original study and indicates that 10 mg/kg of AOM should not cause substantial mortality rate; in contrast, AOM administration has resulted in a significantly higher mortality rate in this replication study, which could be due to the differences in AOM lots, mouse facilities, microbiota, etc. It appears reasonable that the difference in the effects of NC101 to NC101 *Δpks* on the intestinal inflammation and tumorigenesis, as reported in the original study, could not be replicated, with the more severe disease development in this replication study. However, in order to firmly evaluate the reproducibility of the original study, especially the effects of NC101 to NC101 *Δpks* on the intestinal inflammation and tumorigenesis, the experimental conditions (such as the administrated AOM dosage) could be adjusted in this replication study to get enough animals surviving through 18 weeks for the further analyses.

*Reviewer #4:*

This current study is a replication of Arthur et al., 2012. The current study was able to reproduce many findings of the original 2012 study: no differences in bacterial growth and colonization between strains, no difference in histologic inflammation *in vivo*, no difference in detecting the pks island by PCR and sequencing. The authors also include a meta-analysis of the current and former studies. However, the main finding that *E. coli* pks promotes cancer could not be reproduced. Significant differences in experimental timeframe likely underlie this observation – see below. I also have a significant concern about mention of "bacterial contamination" and the extent of AOM toxicity.

1) In both studies, colonization was established for 4 weeks, after which AOM was administered for 6 weeks. However, the current study evaluated survival and tumorigenesis 18 weeks after the last (6th) AOM injection, whereas the original 2012 study evaluated survival and tumorigenesis 18 weeks after colonization. This equates to 10 weeks longer in the current study, which is 50% greater time than in the original, and could explain why no difference in tumorigenesis was observed. The authors in fact indicate that it was difficult to detect differences in tumorigenesis because it had advanced so far and many mice did not survive to this time point. This substantial difference in experimental timing – 18 weeks (2012 study) and 28 weeks (current study) – is a major flaw in this replication study. This difference should be clearly stated in the Abstract and manuscript text. Further, I”m not sure if the meta-analysis in Figure 4 is appropriate with such differences in time frame between the two studies.

2) Potential AOM toxicity: It is concerning that AOM treatment killed 30-70% of the mice, simply during the 6 week injection period. I am not aware of this extent of toxicity in AOM/DSS or AOM/*Il10^-/-^* studies. Perhaps the current authors injected mice with a very concentrated AOM solution? Survival curves from the original 2012 paper do not indicate this extent of toxicity. I recommend the authors amend their Abstract and manuscript text to indicate how different these results were between the current and former studies.

3) Figure 2 figure legend mentions "animals where bacterial contamination was detected (7 out of 84) were censored". What does this mean? Does this mean that some animals housed in gnotobiotic (mono-associated) isolators became contaminated? This is highly concerning. It also raises the possibility that these mice were also infected with other non-bacterial microorganisms, such as virus or fungus. Any contamination could alter results. Potential contamination is only mentioned in this figure legend and does not appear to be mentioned in Results section or Materials and methods section. The authors must explain in greater detail what this "bacterial contamination" means – both to this reviewer and to the readers.

[Editors” note: further revisions were requested prior to acceptance, as described below.]

Thank you for resubmitting your work entitled "Replication Study: Intestinal inflammation targets cancer-inducing activity of the microbiota" for further consideration at *eLife*. Your revised article has been favorably evaluated by Wendy Garrett (Senior Editor) and a Reviewing Editor.

The manuscript has been improved but there are some remaining issues that need to be addressed before acceptance, as outlined below:

The authors provided a highly responsive revision of the original paper. In re-reviewing Eaton, 2015 (study plan) and the original Arthur et al., paper (2012), we agree that the time line for the mouse experiments proposed in Eaton, 2015 was clear whereas the time line between bacterial inoculation, AOM administration and mouse harvest was hard to discern in the original paper. Besides the importance of clearer time line delineation in original manuscripts, this highlights flaws in the review of the original study plan, certainly a distributed responsibility in which several reviews failed to raise any questions.

Thus, in the review of the current manuscript, there a number of places where the text should be modified to be clearer and more detailed as listed below.

1) Introduction. The authors state incorrectly that in Arthur et al., a 100-fold increase in *E. coli* NC101 was detected in the lumen of Il10 KO mice relative to WT controls. This is Figure 1I in Arthur et al. In this experiment, conducted at 20 weeks after transfer to SPF conditions, only the presence of *E. coli* was examined, not NC101 which would have required specific testing for pks or colibactin. NC101 is referred to but, not specifically tested for at this juncture in the Arthur et al., paper.

2) Subsection “Intestinal tumorigenesis and inflammation of germ-free *Il10^-/-^* mice mono-associated with *E. coli* NC101 or NC101𝛥pks”. It was difficult to find wording in the Arthur et al. paper that clearly delineated whether harvest occurred 18 weeks from monoassociation vs 18 weeks post-AOM (wording was “14 and 18 weeks with AOM”) and better definition was missed by several reviews of the Eaton et al. plan including, even perhaps the original authors, as such the following wording change is suggested: “based on methods derived from the original study and not corrected on review of Eaton et al.” as this may represent a more accurate presentation of what occurred.

3) Need to insert the starting N of mice (NC101 39, NC101*Δpks* 45 mice) since the reader can‘t interpret the mouse survival numbers without this information.

4 Please add the range of days of survival to augment the median survival stated.

5) Please expand “suggesting bacterial load was not a factor in the survival differences” as follows: “suggesting while bacterial load was not a factor in the survival differences, host:bacterial interactions could have contribute to the early demise” (or similar).

6) Why are half the mice surviving in the NC101*Δpks* group vs the NC101 group considered a “small impact”? The 57 day survival is the NC101*Δpks* group but it is stated to the NC101 group and then median survival is stated as not able to be calculated but provided as 154 days previously. Please clarify for the reader.

7) The timing of the onset of colitis is unclear. Do the authors mean by 4 weeks of monocolonization or only following AOM?

8) What was the timing of detection of the anal carcinoma?

9) The mouse colon is well known to have a distal “smooth” part grossly and then the proximal half has a “feathered” gross structure. Please provide clarity on whether distal colon refers to the smooth section of the mouse colon or not. For example, in the enterotoxigenic B fragilis model of colon tumorigenesis, the transition between the smooth and feathered colon in the most severely affected animals is essentially a “hard stop” for further tumorigenesis (i.e., tumors push to this transition zone and then rarely penetrate into the feathered area of the mouse colon). Is that is what is being described here?

10) Please clarify the statement: “These observations were the same whether the mouse was mono-associated with NC101 or NC101*Δpks*”. Was this true even when mice were harvested at a time point close to the time that mice were harvested in the original Arthur et al. paper (i.e., day ~126)? The tumorigenesis results of any mice harvested near the timing of the original Arthur et al. paper should be commented on specifically. It seems possible that the time line in this model is crucial for differentiating the outcomes of NC101 and NC101*Δpks*.

11) What is the timing of the pictures in Figure 3—figure supplement 1?

12) Same question as 10 above. Are there any mice analyzed at the time point of the harvests in the Arthur et al., paper?

13) Suggested edit for clarity: change “14 weeks post-AOM treatment” to “14 weeks post-AOM treatment (5 weeks beyond the endpoint in Arthur et al.,)”.

14) Are the median scores reported correct? They have exactly the same numbers and statistical significance seems doubtful as reported. This requires either more explanation or correction.

15) Suggest adding “more severe and/or progressive over time” to the sentence “The absolute scores were greater in this replication attempt compared to the original study, particularly for inflammation and invasion, which, combined with the survival and histopathological observations described above suggests that lesions were more severe in this replication attempt than in the original study.”

16) Suggest adding after “AOM treatment” the following sentence: These results highlight the importance of experimental time lines in assessing differences in mouse models and between reports.” (or similar).

17) “[…], had a natural tendency to have a reduced inflammatory response and tumor load compared to ones that died, then the data reported would be distorted”: The word “distorted” does not seem apt. Mouse models and mice within experiments can be remarkably variable including littermates, mice caged together etc. Isn‘t this the nature of mouse experiments and the point is that investigators need to be aware that inbred mice by no means provide clear “smoothing” of the data?

18) “Additionally, under these conditions it is possible the products of excessive inflammatory responses (e.g. reactive oxygen species”: It would be clearer to state: “Additionally over time […]”.

19) “This increased severity confounds the ability to detect differences […]”: Again, suggest changing wording to “increased severity over time confounds […]”

20) Subsection “Meta-analyses of original and replicated effects”. Again, for clarity change “a common effect size was calculated for each effect from the original and replication studies” to “calculated for survival [.,,]”.

21) Subsection “Genome sequencing data processing and assembly”. Both NC101 and NC101*Δpks* were sequenced. Are these data submitted to GenBank? The accession numbers should be provided.

22) Subsection “Determination of *E. coli* CFU”. What does “intestinal tissue feces” mean? Do the authors mean that intestinal luminal contents were removed from the colon at the time of mouse necropsy? Please clarify.

23) Subsection “Statistical Analysis”. Suggest “reported in the original study in Supplemental […]”.

24) Figure 1A,B should be revised to include much more data that would encapsulate the experiments and their contrast for the reader. Suggest adding numbers of mice for A,B experiments including number of males, females; mark the timing of mouse deaths with arrows; mark timing of onset of detection of colitis and tumors/invasive cancers (per the text the timing of onset of detection of carcinoma was at/near the time point at which the Arthur et al., experiments ended); add to legend what the cross lines added to the timeline represent (mouse censoring due to bacterial contamination). Ideally this figure would enable the reader to readily capture the contrasts between the studies and their timelines. Hopefully any needed information from Arthur et al. would be available to complete this.

25) Figure 2 legend. (n=7 in 4 cages), correct?

26) Figure 2—figure supplement 1. Add range of days/wks that sacrifices occurred.

27) Figure 3—figure supplement 2 legend. Please add the timing of harvest of each mouse displayed in B,C, D and E, ideally on the figure. In Figure 3E, the herniation described is not really visible to the reader. Please provide a higher power inset. Similarly, please clarify Figure 3—figure supplement and Figure 3. Are these sections all from one mouse or different mice? What is the timing of the necropsies leading to these images? Sizing bars are missing from images A and C. In A, the “mucus lakes” should be marked and likely a higher power image of these provided. C appears to be at higher magnification than B. It would be best to show B and C at the same magnification.

[Editors” note: further revisions were requested prior to acceptance, as described below.]

Thank you for resubmitting your work entitled "Replication Study: Intestinal inflammation targets cancer-inducing activity of the microbiota" for further consideration at *eLife*. Your revised article has been favorably evaluated by Wendy Garrett (Senior Editor) and a Reviewing Editor.

The manuscript has been improved and a highly responsive revision is again noted. However, there are some remaining issues that need to be addressed before acceptance, as outlined below:

Two data concerns, both seem like typos:

1) Subsection “Intestinal tumorigenesis and inflammation of germ-free *Il10^-/-^* mice mono-associated with *E. coli* NC101 or NC101𝛥pks”. Looking at Figure 2B, this reviewing editor thinks that NC101 and NC101*Δpks* are reversed. Namely, NC101(not NC101*Δpks*) median survival cannot be determined because more than 1/2 the mice were alive at 18 weeks following monoassociation.

2) Subsection “Intestinal tumorigenesis and inflammation of germ-free *Il10^-/-^* mice mono-associated with *E. coli* NC101 or NC101𝛥pks”. This reviewer thinks the authors are repeating the data although the language differs (lesions and macroscopic tumor burden). The authors note that a “statistically significant decrease in neoplastic lesions” reported in Arthur et al., but provide identical median scores. Please clarify.

---

## [Author Response]

[…] Essential revisions:As pointed out by all reviewers, the attempt at replication primarily revealed the technical aspects that must be controlled for in working with this mouse model in future experiments. A careful standardization of the protocol/disease course is required in individual laboratories before comparing different E. coli strains. Key features of the model to be verified before strain testing include low mortality rates, distinguishable individual tumors at the endpoint and additional controls including non-colonized AOM-injected-only control mice and mono-colonization-only and possibly AOM-untreated control mice. Revising the paper to focus on the technical limitations rather than that the replication attempt led to results differing from the original paper seems prudent. Ultimately, since the biology and kinetics of the Arthur et al., model were not faithfully replicated, these data are unable to address whether, under the conditions of the original paper, NC101 and ΔPKS NC101 differ in inflammation, invasion and neoplasia. This final point should be made clear to the readers of the paper.

Thank you for sharing this critical information. We were unaware of the differences in experimental timing supplied by reviewer 4. As raised by reviewer 1 the original paper was not well described in terms of the timepoints. The experimental timing was based on the information in the original paper and described in the Registered Report with the mice to be sacrificed “18 weeks after last AOM injection”. This remained after informal review and feedback by the original authors during preparation of the Registered Report manuscript, peer review of the Registered Report, and post-publication peer review of the published Registered Report. This was also not raised in the other independent reviews of this Replication Study manuscript. One approach to mitigate the potential for misinterpreting complex study designs is to include a timeline diagram or flowchart as recommended by the ARRIVE Guidelines. We included these points in the revised manuscript.

We also agree with the reviewer regarding the implications this has on the presentation and interpretation of the replication data. We have revised the figures, manuscript, and the Abstract to reflect the difference in experimental timing between the original study and this replication attempt. This includes removing the histopathological analysis from the meta-analysis and revising the survival meta-analysis to reflect the shared time frame between the two studies. We also revised the manuscript regarding the implications this has on the presentation and interpretation of the replication data.

Reviewer #1:[…] Specific comments:1) Mortality seen on Figure 2A. These curves may represent two different types of mortalities (causes)- AOM induced mortality early on and then inflammation and/or tumor load induced mortality later.Early mortality in the Replication experiment is clearly higher than in Arthur et al. (>50% vs less than 30%).What is presented on Figure 2B may be very similar to the data on Supplementary Figure 10 by Arthur et al., (although original paper mortality curve is not well described in terms of what timepoints there are plotted), if mortality between day 0 and 10 is still attributed to the action of AOM. Then long time there is no mortality (days 20 to 80) in both Replication experiment and in Arthur et al. Then mice in Arthur et al., seemingly start to die, but they are immediately collected (all of them), and experiment stopped, so we do not know what happens to mortality. In Replication experiment past day 90 mortality starts but mice are euthanized according to symptoms or dying, so there is an impression of high mortality.

We have revised this figure, the manuscript, and the Abstract to reflect the difference in experimental timing between the original study and this replication attempt in light of the information supplied by reviewer 4.

2) One important discrepancy here is that in Arthur et al., 14 wk and 18 wk point mice were collected when they still were not dying, indicating that probably there is still a room for a difference in inflammation and tumor load/invasion.In Replication attempt, it seems from the Figure legend that mice were mostly collected and "declared dead" when they were already sick (right thing to do in terms of following IACUC regulation) but one could think that by the time PKS+ and ΔPKS mice are already equally very sick, tumor load and inflammation may be already too high and indistinguishable. Indeed, authors note that overall role of inflammation, tumor load and mortality in their experiment was higher than in initial experiment.

We have revised the figures, the manuscript, and the Abstract to reflect the difference in experimental timing between the original study and this replication attempt in light of the information supplied by reviewer 4.

3) Another thing noted during initial planning of experiments and review of Registered Report is that AOM quality may vary. Indeed, different lots of AOM may have different amount of "acting" AOM and sometimes it”s too much- more toxicity/mortality and more tumors in residual mice or poor decayed AOM- more toxicity/mortality and less tumors at the end. Seemingly, in Replication report higher mortality (presumably AOM induced) and higher tumor load is seen.

We agree the observed toxicity is unexpected, especially since the dose and treatment schedule used is below the published toxic dose for AOM in mice. We have revised the manuscript and the Abstract to reflect the difference in experimental timing between the original study and this replication attempt in light of the information supplied by reviewer 4.

4) Since AOM metabolism and potentially bacterial and liver metabolism may be involved, the diets used for germ free mice should be compared and that should be discussed.

We agree that diets could be a factor and have included this in the revised manuscript. We also used the same diet as the original study (as communicated by the original authors) and have included this in the revised manuscript.

5) Using Chi- values for tumor numbers in Replication study when tumors cannot be enumerated (why?) further complicates the interpretation.

We revised the figure to better distinguish the tumors that could not be enumerated from the ones that could. This is included in the figure to illustrate the number that could not be quantified in both groups because of the coalescing nature of the lesions.

6) Another important caveat which should be acknowledged also stems from high mortality. When mortality is that high, there may be a natural selection for mice with lower inflammatory response and lower tumor load- and the mice where potentially there was (could be) a difference have been kept for too long under these conditions of inflammation and AOM lot. This is not very scientific argument, I know, but after this Replication attempt is published, the field will need all of the possible scenarios/explanations discussed.

We agree that the increased severity in the replication attempt complicates the interpretation of the data. Besides leading to information loss, if the mice that survived, and thus were quantified for inflammation and tumorigenesis, had a natural tendency to have a reduced inflammatory response and tumor load compared to ones that died, then the data reported would be distorted, limiting the opportunity to detect differences between the two groups. We have included this point in the revised manuscript.

7) Inflammation is known to promote cancer. PKS+ bacteria is proposed to be more carcinogenic than PKSΔ. However, under the conditions of stronger inflammation (observed in Replication attempt), Inflammation arguably can become more important than PKS status. PKS is genotoxic, but so are the products of excessive inflammatory responses (ROS, RNI etc.). One could argue that under stronger inflammation the presence or absence of PKS becomes less relevant as for its effect towards tumor progression. Therefore, comparison of this (Replication) and Arthur et al. studies with regard to tumor characteristics only makes sense if the same levels of inflammation are achieved.

We have revised the figures, the manuscript, and the Abstract to reflect the difference in experimental timing between the original study and this replication attempt in light of the information supplied by reviewer 4.

In my view, for these experiments the technical attempt to reproduce the results has been attempted but based on the confounded data it is difficult to judge whether important conceptual results are reproduced or not (i.e. whether PKS+ is superior at tumor progression).The only thing which can be concluded, in my view, is that during the Replication attempt, the team did not achieve the same levels of AOM toxicity/action, the same levels of inflammation and tumor load (more); and observed higher mortality. Under these circumstances it is still not clear whether PKS+ E. coli induces more tumors and more aggressive tumors; than ΔPKS controls.

We have revised the figures, the manuscript, and the Abstract to reflect the difference in experimental timing between the original study and this replication attempt in light of the information supplied by reviewer 4.

Reviewer #2:[…] Major points:The major aim of the present study is to compare the potential of the E. coli strains NC101 and NC101𝛥pks in inducing intestinal inflammation and tumor formation in monocolonized GF Il10^-/-^ mice in order to evaluate the reproducibility of the experiments published by Arthur et al. For this comparison to be performed the kinetics of the model should be similar to what was reported by Arthur et al. In the present study mice mono-colonized with both E. coli strains display a much higher lethality and an exacerbated phenotype by histopathology as compared to the report by Arthur et al. A very low number of mice met the endpoint of the study. Because of these issues no conclusions can be drawn regarding the potential of E. coli NC101 vs NC101𝛥pks to induce intestinal inflammation and tumor formation in monocolonized GF Il10^-/-^ mice.

We have revised the figures, the manuscript, and the Abstract to reflect the difference in experimental timing between the original study and this replication attempt in light of the information supplied by reviewer 4.

In the Abstract the authors first mention that "Mono-association […] resulted in similar levels of intestinal inflammation and tumorigenesis; whereas the original study reported decreased tumor multiplicity […]" and secondly, they mention that this replication study showed more severe histopathological observations and a much lower survival rate compared to the original study. This structure of the Abstract is misleading to the reader. The strong differences in the kinetics of the experiment compared to the original report and the lack of power of the current study to assess the reproducibility of the original findings should be mentioned first. The observations regarding tumor number and histopathological features could be discussed as indicative of a lack of difference in the context of a severely exacerbated disease course.

We have revised the Abstract to reflect the difference in experimental timing between the original study and this replication attempt in light of the information supplied by reviewer 4.

In my opinion, the major conclusion of this study is that the AOM/E. coli NC101 Il10^-/-^ mono-colonization model is characterized by important technical limitations that should be carefully controlled in future studies. A careful standardization of the protocol/disease course is required in individual laboratories before comparing different E. coli strains. This should lead to low mortality rates and distinguishable individual tumors at the endpoint. Including non-colonized AOM-injected-only control mice and mono-colonization-only, AOM-untreated control mice is essential for the correct interpretation of results in this model. Such factors should be considered for the experimental design of future studies.

We agree. Our findings clearly demonstrate the essential nature of clear and precise reporting of experimental details to ensure published research are accurately compared, reproduced, and interpreted. We have included these factors as considerations for the experimental design of future studies in the revised manuscript.

Additional points:The comparison of the death rates between this study and Arthur et al. is vague in this part of the manuscript despite the fact that this is the major limitation of this replication attempt. The authors should clearly mention the percentages of mice meeting the endpoint in the two studies.

We have revised the figures, the manuscript, and the abstract to reflect the difference in experimental timing between the original study and this replication attempt in light of the information supplied by reviewer 4. We have also moved the percentages of mice earlier in the text, when comparing death rates.

Mortality was highest during AOM treatment independently of the bacterial strain used whereas mice started dying right after the first AOM injection. The interpretation of these observations is difficult because of the absence of AOM-only controls: lethality over the injection period may be an effect of AOM toxicity alone or of AOM in combination with the effect of E.coli. It may also be an effect of the injection process itself (bleeding of internal organs for example). This is an additional technical variable that should be mentioned in the Discussion.

We agree and have included these factors as considerations for the experimental design of future studies in the revised manuscript.

The authors mention that mortality during AOM injections was greater in mice colonized with NC101Δpks. Is this difference statistically significant? This should be mentioned and if significant further discussed.

We have revised the figures, the manuscript, and the abstract to reflect the difference in experimental timing between the original study and this replication attempt in light of the information supplied by reviewer 4. Further discussion for the increased mortality during AOM injections was included

The authors discuss sex-related differences which may confound the differences observed by Arthur et al. where 4 female and 8 male mice mono-associated with NC101 and 8 male mice mono-associated with NC101Δpks were analyzed. What is the result of these analyses if only male mice are considered? (n = 8 vs 8)?

This is an interesting point. During preparation of the Registered Report we were informed of the ratio of female and male mice in their experiment; however, we did not receive the raw data from the authors that would allow us to conduct this exploratory analysis.

The study is not powered enough to draw conclusions from the histopathological analyses because of the small number of mice resulting from an exacerbated disease course in the experiment.

We agree we did not achieve the target number of 20 (instead reaching 10), which was based off the data from the 14 mice reported in the original study (at the 18 week timepoint). However, we have removed all analysis from the histopathological analysis in light of the information supplied by reviewer 4.

Figure 2A: Log-rank test not shown in the legend.

We included this exploratory test (of the entire timecourse) in the figure legend and manuscript in addition to the revised analysis of the comparable timecourse between the original study and this replication in light of the information supplied by reviewer 4.

Reviewer #3:[…] It appears that the authors of this replication study had tried the best to conduct the experiments under the conditions as close as possible to the original study. For instance, bacterial strains were shared by the Arthur Lab (subsection “Bacterial strains and growth conditions”), germ-free Il10^-/-^ mice were derived from the same germ-free colony used in the original study) (subsection “Intestinal tumorigenesis and inflammation of germ-free Il10^-/-^ mice mono-associated with E. 121 coli NC101 or NC101𝛥pks”), and the used AOM dose, 10 mg/kg, was the same as the original study (the same vendor, same catalog number, but with different lot number). AOM administration (the same vendor, same catalog number, same dosage used, but with different lot number) had resulted in a significantly higher mortality rate in this replication study compared with the original study. As cited by the authors (subsection “Intestinal tumorigenesis and inflammation of germ-free Il10^-/-^ mice mono-associated with E. 121 coli NC101 or NC101𝛥pks”), a previous study (Bissahoyo, 2005) showed that 33 mice were treated with 10 mg/kg of AOM, and "no premature loss of mice" was observed at "six months after the first AOM dose". This appears in line with the original study and indicates that 10 mg/kg of AOM should not cause substantial mortality rate; in contrast, AOM administration has resulted in a significantly higher mortality rate in this replication study, which could be due to the differences in AOM lots, mouse facilities, microbiota, etc. It appears reasonable that the difference in the effects of NC101 to NC101 Δpks on the intestinal inflammation and tumorigenesis, as reported in the original study, could not be replicated, with the more severe disease development in this replication study. However, in order to firmly evaluate the reproducibility of the original study, especially the effects of NC101 to NC101 Δpks on the intestinal inflammation and tumorigenesis, the experimental conditions (such as the administrated AOM dosage) could be adjusted in this replication study to get enough animals surviving through 18 weeks for the further analyses.

We have revised the figures, the manuscript, and the Abstract to reflect the difference in experimental timing between the original study and this replication attempt in light of the information supplied by reviewer 4.

Reviewer #4:[…] Essential revisions:1) In both studies, colonization was established for 4 weeks, after which AOM was administered for 6 weeks. However, the current study evaluated survival and tumorigenesis 18 weeks after the last (6th) AOM injection, whereas the original 2012 study evaluated survival and tumorigenesis 18 weeks after colonization. This equates to 10 weeks longer in the current study, which is 50% greater time than in the original, and could explain why no difference in tumorigenesis was observed. The authors in fact indicate that it was difficult to detect differences in tumorigenesis because it had advanced so far and many mice did not survive to this time point. This substantial difference in experimental timing – 18 weeks (2012 study) and 28 weeks (current study) – is a major flaw in this replication study. This difference should be clearly stated in the Abstract and manuscript text. Further, I”m not sure if the meta-analysis in Figure 4 is appropriate with such differences in time frame between the two studies.

Thank you for sharing this critical information. We were unaware of these differences in experimental timing. As raised by reviewer 1 the original paper was not well described in terms of the timepoints. The experimental timing was based on the information in the original paper and described in the Registered Report with the mice to be sacrificed “18 weeks after last AOM injection”. This remained after informal review and feedback by the original authors during preparation of the Registered Report manuscript, peer review of the Registered Report, and post-publication peer review of the published Registered Report. This was also not raised in the other independent reviews of this Replication Study manuscript. One approach to mitigate the potential for misinterpreting complex study designs is to include a timeline diagram or flowchart as recommended by the ARRIVE Guidelines. We included these points in the revised manuscript.

We also agree with the reviewer regarding the implications this has on the presentation and interpretation of the replication data. We have revised the figures, manuscript, and the Abstract to reflect the difference in experimental timing between the original study and this replication attempt. This includes removing the histopathological analysis from the meta-analysis and revising the survival meta-analysis to reflect the shared time frame between the two studies.

2) Potential AOM toxicity: It is concerning that AOM treatment killed 30-70% of the mice, simply during the 6 week injection period. I am not aware of this extent of toxicity in AOM/DSS or AOM/Il10^-/-^ studies. Perhaps the current authors injected mice with a very concentrated AOM solution? Survival curves from the original 2012 paper do not indicate this extent of toxicity. I recommend the authors amend their Abstract and manuscript text to indicate how different these results were between the current and former studies.

We agree the observed toxicity is unexpected, especially since the dose and treatment schedule used is below the published toxic dose for AOM in mice and, to our knowledge, is the same used in the original study. The cause of early death following AOM injection was confirmed to be acute liver failure, based on histologic evaluation. Inclusion of AOM-only controls could have indicated any increased susceptibility to AOM of the mice used in this study. We have also revised the Abstract to reflect the increased severity between the two studies during AOM treatment.

3) Figure 2 figure legend mentions "animals where bacterial contamination was detected (7 out of 84) were censored". What does this mean? Does this mean that some animals housed in gnotobiotic (mono-associated) isolators became contaminated? This is highly concerning. It also raises the possibility that these mice were also infected with other non-bacterial microorganisms, such as virus or fungus. Any contamination could alter results. Potential contamination is only mentioned in this figure legend and does not appear to be mentioned in Results section or Materials and methods section. The authors must explain in greater detail what this "bacterial contamination" means – both to this reviewer and to the readers.

We have revised the manuscript (Materials and methods section and Figure 2 figure legend) to provide a clearer description of the contaminated mice. In brief, (1) the mice were in isocages, not isolators, with only a few isocages contaminated during the course of the experiment, and (2) the rest of the isocages remained gnotobiotic.

[Editors” note: further revisions were requested prior to acceptance, as described below.]

The authors provided a highly responsive revision of the original paper. In re-reviewing Eaton, 2015 (study plan) and the original Arthur et al., paper (2012), we agree that the time line for the mouse experiments proposed in Eaton, 2015 was clear whereas the time line between bacterial inoculation, AOM administration and mouse harvest was hard to discern in the original paper. Besides the importance of clearer time line delineation in original manuscripts, this highlights flaws in the review of the original study plan, certainly a distributed responsibility in which several reviews failed to raise any questions.Thus, in the review of the current manuscript, there a number of places where the text should be modified to be clearer and more detailed as listed below.1) Introduction. The authors state incorrectly that in Arthur et al. a 100-fold increase in E. coli NC101 was detected in the lumen of Il10 KO mice relative to WT controls. This is Figure 1I in Arthur et al. In this experiment, conducted at 20 weeks after transfer to SPF conditions, only the presence of E. coli was examined, not NC101 which would have required specific testing for pks or colibactin. NC101 is referred to but, not specifically tested for at this juncture in the Arthur et al., paper.

Thank you for raising this error. We have revised the manuscript to remove this statement.

2) Subsection “Intestinal tumorigenesis and inflammation of germ-free Il10^-/-^ mice mono-associated with E. coli NC101 or NC101𝛥pks”. It was difficult to find wording in the Arthur et al. paper that clearly delineated whether harvest occurred 18 weeks from monoassociation vs 18 weeks post-AOM (wording was “14 and 18 weeks with AOM”) and better definition was missed by several reviews of the Eaton et al. plan including, even perhaps the original authors, as such the following wording change is suggested: “based on methods derived from the original study and not corrected on review of Eaton et al. As this may represent a more accurate presentation of what occurred.

We agree and have revised this sentence as suggested.

3) Subsection “Intestinal tumorigenesis and inflammation of germ-free Il10^-/-^ mice mono-associated with E. coli NC101 or NC101𝛥pks”. Need to insert here the starting N of mice (NC101 39, NC101Δpks 45 mice) since the reader can’t interpret the mouse survival numbers without this information.

We agree and have included this in the revised manuscript.

4) Please add the range of days of survival to augment the median survival stated.

We have included the range of survival times for each group.

*5)* Please expand “suggesting bacterial load was not a factor in the survival differences” as follows: “suggesting while bacterial load was not a factor in the survival differences, host:bacterial interactions could have contribute to the early demise” (or similar).

While the interpretation that bacterial load was not a factor, it is not clear that host:bacterial interactions necessarily contributed. The early death rate in response to AOM was not explained in this study and thus we feel it is not appropriate to speculate.

6) There to be an error. Why are half the mice surviving in the NC101Δpks group vs the NC101 group considered a “small impact”? The 57 day survival is the NC101Δpks group but it is stated to the NC101 group and then median survival is stated as not able to be calculated but provided as 154 days previously. Please clarify for the reader.

This analysis was to compare the replication results to the original study. In order to do that we treated 18 weeks after mono-association as the study end point (i.e. ignoring all events after this time point). We have revised the manuscript to better clarify what these results represent. We have also removed the word “small” as this was indeed an error.

7) The timing of the onset of colitis is unclear. Do the authors mean by 4 weeks of monocolonization or only following AOM?

We have revised the manuscript to reflect the timing of the onset of colitis. Specifically, the first mouse was at 5 weeks after monocolonization.

8) What was the timing of detection of the anal carcinoma?

We have included the timing of the anal carcinoma in the revised manuscript. Specifically, this was observed between 19 and 27 weeks after monocolonization.

9) The mouse colon is well known to have a distal “smooth” part grossly and then the proximal half has a “feathered” gross structure. Please provide clarity on whether distal colon refers to the smooth section of the mouse colon or not. For example, in the enterotoxigenic B fragilis model of colon tumorigenesis, the transition between the smooth and feathered colon in the most severely affected animals is essentially a “hard stop” for further tumorigenesis (i.e., tumors push to this transition zone and then rarely penetrate into the feathered area of the mouse colon). Is that what is being described here?

We have revised this section to better clarify the observations. The most extensive tumors reached approximately mid-colon. The gross appearance of the proximal mucosa was not recorded.

10) Please clarify the statement: “These observations were the same whether the mouse was mono-associated with NC101 or NC101Δpks”. Was this true even when mice were harvested at a time point close to the time that mice were harvested in the original Arthur et al. paper (i.e., day ~126)? The tumorigenesis results of any mice harvested near the timing of the original Arthur et al. paper should be commented on specifically. It seems possible that the time line in this model is crucial for differentiating the outcomes of NC101 and NC101Δpks.

We have revised this section to clarify the range these observations were made. We also included in a previous paragraph the observations of the mice that were harvested at times close to the original study.

11) What is the timing of the pictures in Figure 3—figure supplement 1?

This is from a mouse that was mono-associated with NC101 and died 13 weeks after AOM treatment. The timing is described in the figure legend.

12) Are there any mice analyzed at the time point of the harvests in the Arthur et al., paper?

We included the available information in the paragraph included in response to point 10 above.

*13)* Suggested edit for clarity: change “14 weeks post-AOM treatment” to “14 weeks post-AOM treatment (5 weeks beyond the endpoint in Arthur et al.,)”.

We agree and included this in the revised manuscript.

14) Subsection “Intestinal tumorigenesis and inflammation of germ-free Il10^-/-^ mice mono-associated with E. coli NC101 or NC101𝛥pks”. Are these scores correct? They have exactly the same numbers and statistical significance seems doubtful as reported. This requires either more explanation or correction.

These are the median scores reported in the original study. The original study analyzed the ordinal scoring data as interval measurements (by *t* test), which is not appropriate since the mean cannot be defined (Baker et al., 2014; Gibson-Corley et al., 2013). That is, while a number is used for the scoring it represents non-numeric concepts like “severe or high grade dysplasia characterized as adenoma, restricted to the mucosa”. We conducted a non-parametric test (i.e. Mann Whitney test) on the original data which gave similar results. We included this additional explanation in the revised manuscript.

15) Suggest adding “more severe and/or progressive over time” to the sentence “The absolute scores were greater in this replication attempt compared to the original study, particularly for inflammation and invasion, which, combined with the survival and histopathological observations described above suggests that lesions were more severe in this replication attempt than in the original study.”

We agree and included this in the revised manuscript. We have also revised this section to better clarify the observations in light of the differences in methodology between the two studies.

16) Subsection “Intestinal tumorigenesis and inflammation of germ-free Il10^-/-^ mice mono-associated with E. coli NC101 or NC101𝛥pks”. Suggest adding after “AOM treatment” the following sentence: These results highlight the importance of experimental time lines in assessing differences in mouse models and between reports.” (or similar).

We have revised this section to highlight that it is possible that the experimental timing in this mouse model is crucial for differentiating the outcomes of NC101 and NC101Δ*pks*. Additionally, we have included a sentence that methods should be clearly described and published to facilitate reproducibility.

17) “[…], had a natural tendency to have a reduced inflammatory response and tumor load compared to ones that died, then the data reported would be distorted”: The word “distorted” does not seem apt. Mouse models and mice within experiments can be remarkably variable including littermates, mice caged together etc. Isn’t this the nature of mouse experiments and the point is that investigators need to be aware that inbred mice by no means provide clear “smoothing” of the data?

We removed this sentence and revised this section to better clarify the observations in light of the differences in methodology between the two studies.

18) “Additionally, under these conditions it is possible the products of excessive inflammatory responses (e.g. reactive oxygen species”: It would be clearer to state: “Additionally over time […]”.

We agree and included this in the revised manuscript.

19) “This increased severity confounds the ability to detect differences […]”: Again, suggest changing wording to “increased severity over time confounds […]”

We removed this sentence and revised this section to better clarify the observations in light of the differences in methodology between the two studies.

20) Subsection “Meta-analyses of original and replicated effects”. Again, for clarity change “a common effect size was calculated for each effect from the original and replication studies” to “calculated for survival [.,,]”.

We agree and included this in the revised manuscript.

21) Subsection “Genome sequencing data processing and assembly”. Both NC101 and NC101Δpks were sequenced. Are these data submitted to GenBank? The accession numbers should be provided.

We have included the accession numbers in the revised manuscript.

22) Subsection “Determination of E. coli CFU”. What does “intestinal tissue feces” mean? Do the authors mean that intestinal luminal contents were removed from the colon at the time of mouse necropsy? Please clarify.

This was a typographic error. “Intestinal tissue” has been deleted.

23) Subsection “Statistical Analysis”. Suggest “reported in the original study in Supplemental […]”.

We agree and included this in the revised manuscript.

24) Figure 1A,B should be revised to include much more data that would encapsulate the experiments and their contrast for the reader. Suggest adding numbers of mice for A,B experiments including number of males, females; mark the timing of mouse deaths with arrows; mark timing of onset of detection of colitis and tumors/invasive cancers (per the text the timing of onset of detection of carcinoma was at/near the time point at which the Arthur et al., experiments ended); add to legend what the cross lines added to the timeline represent (mouse censoring due to bacterial contamination). Ideally this figure would enable the reader to readily capture the contrasts between the studies and their timelines. Hopefully any needed information from Arthur et al. would be available to complete this.1

We have revised Figure 2A,B to include more information as suggested and included what information was available from Arthur et al. The timing of mouse deaths is already indicated in the Kaplan-Meier plot in the vertical lines, thus we did not include additional marks.

25) Figure 2 legend. (n=7 in 4 cages), correct?

There were only 3 cages. We have revised the figure legend to accurate reflect this in addition to the number of mice (n=7).

26) Figure 2—figure supplement 1. Add range of days/wks that sacrifices occurred.

We have included the range of sacrifices.

27) Figure 3—figure supplement 2 legend. Please add the timing of harvest of each mouse displayed in B,C, D and E, ideally on the figure. In Figure 3E, the herniation described is not really visible to the reader. Please provide a higher power inset. Similarly, please clarify Figure 3—figure supplement 3 and Figure 3. Are these sections all from one mouse or different mice? What is the timing of the necropsies leading to these images? Sizing bars are missing from images A and C. In A, the “mucus lakes” should be marked and likely a higher power image of these provided. C appears to be at higher magnification than B. It would be best to show B and C at the same magnification.

The time of euthanasia has been added to the figure legends. The herniated gland in Figure 3—figure supplement 2E has been circled to clarify it. Size bars have been added to Figure 3—figure supplement 3. Figure 3—figure supplement B,C are at different magnifications because they demonstrate different distributions of lesions. Parts A and C are from the same mouse, while part B is from a different mouse.

[Editors” note: further revisions were requested prior to acceptance, as described below.]

The manuscript has been improved and a highly responsive revision is again noted. However, there are some remaining issues that need to be addressed before acceptance, as outlined below:Two data concerns--both seem like typos:1) Subsection “Intestinal tumorigenesis and inflammation of germ-free Il10^-/-^ mice mono-associated with E. 124 coli NC101 or NC101𝛥pks”. Looking at Figure 2B, this reviewing Editor thinks that NC101 and NC101Δpks are reversed. Namely, NC101(not NC101Δpks) median survival cannot be determined because more than 1/2 the mice were alive at 18 weeks following monoassociation.

Thank you for catching this error. We have corrected it in the revised manuscript.

2) Subsection “Intestinal tumorigenesis and inflammation of germ-free Il10^-/-^ mice mono-associated with E. coliE. coli NC101 or NC101𝛥pks”. This reviewer thinks the authors are repeating the data although the language differs (lesions and macroscopic tumor burden). The authors note that a “statistically significant decrease in neoplastic lesions” reported in Arthur et al., but provide identical median scores. Please clarify.

Thank you for raising these. One reference was to the number of macroscopic tumors, while the other referred to the scoring of tissues for neoplasia. We have revised the latter sentence from “neoplastic lesions” to “neoplasia scores” to avoid any confusion and describe the invasion and inflammation scores in the same manner.

The original median scores for neoplasia, invasion, and inflammation are accurately reported and the reviewer is correct that the median scores for NC101 and NC101*Δpks* are identical (both are 4). While a *t*-test was not appropriate for these data, a *U*-test gave the same results, which we report. The statistical difference is because of the spread of the data, despite identical medians. We have included the range for each measure to help clarify this point.